

# Inference of analytical flow duration curves in Swiss alpine environments

Ana Clara Santos[1,2], Maria Manuela Portela[2], Andrea Rinaldo[1,4], and Bettina Schaefli[1,3]

[1]School of Architecture, Civil and Environmental Engineering (ENAC), École Polytechnique Fédérale de Lausanne (EPFL), Switzerland
[2]Instituto Superior Teécnico, Technical University of Lisbon, Lisbon, Portugal
[3]Faculty of Geosciences and Environment, University of Lausanne
[4]Dipartimento di Ingegneria Civile Edile e Ambientale, Universitá degli studi di Padova

*Correspondence to:* Ana Clara Santos (anaclara.santos@epfl.ch)

**Abstract.** This paper assesses the performance of an analytical modeling framework for streamflow probability distributions for summer streamflow of 26 Swiss catchments characterized by negligible anthropic influence. These catchments show a wide range of hydroclimatic regimes, including snow- and icemelt influenced streamflows. The model parameters are estimated from a gridded daily precipitation data set and observed daily discharge time series. The performance of the linear and nonlinear

5   model version is assessed in terms of reproducing observed flow duration curves and their natural variability. The results show that the model performs well for summer discharges under all analyzed regimes and that there is a clear model performance increase with mean catchment elevation (i.e with transition from rainfall-dominated to snow-influenced regimes). The nonlinear model version outperforms the linear model for all regimes but the performance difference decreases also with mean catchment elevation. Future work will focus on the extension of the modeling framework, addressing snowmelt and snowfall onset.

## 10   1   Introduction

Knowledge of the availability and variability of daily discharges in a given stream section proves useful for many engineering applications (e.g. the design of hydro-power plants or water supply systems), as well as for studies about water quality and allocation, or about stream ecology alterations and sediment transport (Vogel and Fennessey, 1995; Searcy, 1959; Ceola et al., 2010; Basso et al., 2015). For many such applications, knowledge of the probability distribution of daily discharges rather than

15   of the serial structure of their occurrence is sufficient, i.e. daily discharges can be treated as a random variable.

The probability distribution of daily discharges is traditionally not represented as a probability density function (pdf) but in terms of flow duration curves (FDCs) that associate an exceedance probability to each discharge value (Vogel and Fennessey, 1994), which corresponds to the complement of the cumulative distribution function (cdf).

FDCs or other representations of the probabilistic distributions of daily discharges can be obtained in different manners,

20   the most straightforward method being the assignment of empirical probabilities to observed ranked data (yielding empirical FDCs) (Vogel and Fennessey, 1994). Quite frequently, however, there are no discharge data available for the exact location of interest or, if there are, the length of the time series is insufficient, requiring other solutions for FDC estimation.



According to Castellarin et al. (2013), there exist several ways to estimate FDCs in the absence of sufficient streamflow data, including statistical and process-based methods. Statistical methods include regression methods that generate FDCs independently of the characteristics of the catchment and of climate (e.g. Franchini and Suppo (1996)) but generally use geological and topographic characteristics as predictor variables. Geostatistical methods are also relatively popular (Pugliese et al., 2014) to regionalize FDCs according to geographic proximity and topography.

Another group of statistical methods are so-called index flow methods that consist in some form of re-scaling of a reference FDC (Castellarin et al., 2004), which can be obtained by regionalizing the parameters of the reference curve to each location of interest (Ganora et al., 2009) or by using standard representations of FDCs for homogeneous regions and rescaling them to an index-flow ( typically the mean annual runoff or the median daily runoff). Finally, there are also methods that propose to correct a relatively small series of data observations to generate a more reliable FDC using resampling experiments like the one suggested by Castellarin et al. (2004).

Process-based methods, instead, try to combine climate controls and catchment characteristics to estimate the shape of FDCs at ungauged locations. They consist of models that are approximations of the behavior of the given hydrological system, providing mathematical or simulation-based descriptions of FDCs based on observed data other than daily discharges. Simulation-based methods use long-term numerical simulations of the climatic and hydrological processes in a catchment to describe discharge and build FDCs. Being generally relatively complex, simulation-based methods can provide a detailed description of the hydrological system, but are demanding in terms of model development time and necessary data (Viviroli et al., 2009). Derived distribution models, in exchange, derive the FDCs from precipitation and catchment characteristics analytically. One of those models is the one proposed by Botter et al. (2007c) later extended in a variety of ways (Botter et al., 2008, 2009; Schaefli et al., 2013; Müller et al., 2014), that will be explored in this paper.

The discharge in a river section depends on various factors, including climate, geomorphology and land use. Precipitated water can be released and stored in different ways, but the main climate control for discharge production is the precipitation event itself (Wagener et al., 2007), which, in Alpine regions, mostly occurs in form of rain or snow. Botter et al. (2007c) assumed rainfall as the trigger for discharge production and proposed an analytical model to obtain probabilistic distributions of discharge by considering daily discharge as the result of "the superposition of a sequence of subsurface water flow pulses, triggered by a stochastic precipitation and censored by the soil moisture dynamics" (Botter et al., 2013).

Describing rainfall as a stochastic process (a marked Poisson process with exponentially distributed rainfall depths), Botter et al. (2007c) showed that the FDC of rainfall-dominated catchments that present a linear decay of discharge due to the release of water from the subsoil can be described by a gamma distribution characterized by the mean depth of precipitation, the frequency of precipitation events that produce discharge, the area of a catchment and the mean residence time of the catchment. This model framework has been applied successfully in Europe, in Italy (Botter et al., 2007c, 2009; Ceola et al., 2010; Schaefli et al., 2013) and Switzerland (Schaefli et al., 2013; Basso et al., 2015), and in the US (Botter et al., 2007a; Ceola et al., 2010; Botter et al., 2013).

Based on the same theoretical framework, some extensions of the model have already been proposed to make it more suitable for different hydrological conditions, in particular for situations where the decay of discharge due to the release of water from



the soil is considered to be nonlinear (Botter et al., 2009), for winter flow in snow-dominated catchments (Schaefli et al., 2013) and for seasonally dry climates (Müller et al., 2014):

In the previous applications of the model, the focus was generally on the study of signatures of discharge regimes under different climates and landscape conditions (Botter et al., 2007a, 2013), where the shape of the pdf is more important than the

5 accuracy of the obtained discharge probabilities. It is important to emphasize that except for the work of Schaefli et al. (2013) that adapts the model to conditions of snow accumulation, previous applications deliberately exclude catchments and/or seasons where snow accumulation and melt affect the discharges (Botter et al., 2007a; Ceola et al., 2010; Botter et al., 2013; Doulatyari et al., 2015). A detailed analysis of how well the modeling framework can reproduce observed FDCs is presented in the work of Botter et al. (2008) and in particular in the work of Ceola et al. (2010), who presents a detailed analysis of different parameter

estimation methods and of model performance assessment for the linear and the non-linear model version.

The objective of this research is to assess the model performance (linear and non-linear form) for a set of 26 Swiss catchments for summer flows, including rainfall-dominated catchments with typical summer low flows and catchments with snowmelt-driven summer high flows. Particular attention is paid hereby to a detailed analysis of the model parameters and their seasonality.

Switzerland is a mountainous, relatively humid country, with precipitation approximately evenly distributed during the seasons, but varying according to the region (Schwarb et al., 2001), which results in a relatively high diversity of hydrologic regimes within this small country (41.285 km2) (Weingartner and Aschwanden, 1992). Very intense precipitation events are not common, especially not in higher areas, with the exception of the south of Alps region (Spreafico and Weingartner, 2005). These conditions are ideal for the application of the analytical streamflow distribution model of Botter et al. (2007c) because

most of the discharge can be assumed to result from subsurface flow, one of the necessary assumptions underlying this model. This assumption in particular also holds in the presence of snowmelt, which plays a major role in several of the studied catchments even in summer (see Section 3), most of the discharge can be assumed to result from subsurface flow, one of the necessary assumptions underlying this model. It is noteworthy, however, that the travel time distributions underlying the hydrologic response at a chosen control section (stream flow gauge) cannot be considered to be stationary, an issue which is of

major concern when transport processes are investigated (Rinaldo et al., 2011, 2015)

The paper is organized as follows: Section 2 provides a description of the analytical model, together with the methods adopted in this paper to estimate the model parameters and to assess the model performance, followed by a presentation of the Swiss case studies (Section 3). The obtained results for the linear and the nonlinear model version (Section 4) are discussed in Section 5 with a particular focus on the model performance under different hydrological regimes. The conclusions are

30 summarized in Section 6.

## 2 Methods

Hereafter, we first give a short overview over the used analytic model framework, followed by the two different methods used for parameter estimation and for model performance assessment. All model methods are applied only to the summer season

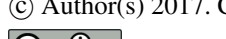



(June $1^{st}$ to August $31^{st}$, see also Section 3). The sequence of calculations required to estimate all model parameters in this work is synthesized in Figure 1.

## 2.1 Model framework

The analytical model framework for probabilistic characterization of rainfall-drive daily discharges developed by Botter et al. (2007c) is based on a previous model proposed by Rodriguez-Iturbe et al. (1999), which model represents the dynamics of soil moisture at a point as a result of a deterministic state-dependent loss function, combined with stochastic increments triggered by rainfall events. Following Botter et al. (2007c), the spatially averaged soil moisture $s(t)$ can be obtained based on the water balance equation:

$$\frac{ds(t)}{dt} = -\rho[s(t)] + \xi_t, \tag{1}$$

where $-\rho[s(t)]$ is the loss function, due to evapotranspiration, surface runoff and deep percolation, and where $\xi_t$ represents the stochastic instantaneous inputs due to rainfall infiltration from rainfall.

Botter et al. (2007c) proposed to describe the dynamics of daily streamflow with a similar stochastic differential equation, supposing that some precipitation events can act as a stochastic forcing for discharge production and that water will be released from the soil producing discharge according to a deterministic decay :

$$\frac{dQ(t)}{dt} = -kQ(t) + \xi_t", \tag{2}$$

where $(Q)$ is the daily streamflow, $k$ is the inverse of the time constant associated with the loss function and $\xi_t"$ is the stochastic rate due to discharge producing precipitation events.

It is assumed hereby that discharge $(Q)$ is the result of a sequence of subsurface inputs triggered by events of precipitation that deliver enough water to fill the water deficit in the soil and to raise its level of moisture above its retention capacity. The excess of water becomes discharge and is removed from the soil as subsurface run-off. As it can be seen in Equation 2, the subsurface storage is assumed to behave like a linear reservoir with a constant $k$. This description comprises only the dynamics of the sub-surface runoff, neglecting any surface runoff.

The overall rainfall forcing can be modeled as a marked Poisson process with frequency $\lambda_P$ and exponentially distributed rainfall depths with average $\alpha$. But not all the rainfall events are able to force discharge production. Accordingly, the frequency is reduced to $\lambda$, the frequency of discharge-producing events, i.e.of events that raise the soil moisture above its retention capacity. In rainfall-driven environments, this reduced frequency $\lambda$ can be understood as the frequency of rainfall events that are unusable by plants; $\lambda$ is influenced by the soil storage capacity and soil drying time (Botter et al., 2007b). The corrected frequency $\lambda$ can be written as (Botter et al., 2007a).

$$\lambda = \eta \frac{\exp(-\gamma)\gamma^{\frac{\lambda_P}{\eta}}}{\Gamma(\lambda_P/\eta, \gamma)}, \tag{3}$$





where $\Gamma(a,b)$ is the lower incomplete Gamma function with parameters $a$ and $b$, $\eta = E/(nZ_r(s_1 - s_w))$, $\gamma = \gamma_P nZ_r(s_1 - s_w)$ and $\gamma_P = 1/\alpha$. $E$ is the maximum evapotranspiration rate and $nZ_r(s_1 - s_w)$ synthesizes the volume in the soil that can be filled by water, where $n$ is the porosity of the soil and $Z_r$ is the effective depth of the soil. The multiplication of these two variables represents a normalizing factor. Accordingly, $s_1$ is the soil moisture threshold above which water starts to move in the soil and $s_w$ is the wilting point. The difference represents the volume in the soil that can be filled before drainage starts.

As discussed in detail by Botter et al. (2007c), this framework results in the following probability distribution of daily discharges at the catchment outlet:

$$p(Q, t \to \infty) = \frac{1}{\Gamma\left(\frac{\lambda}{k}\right)} \frac{1}{Q} \left(\frac{Q}{\alpha k A}\right)^{\frac{\lambda}{k}} \exp\left(-\frac{Q}{\alpha k A}\right), \tag{4}$$

where $Q$ is the daily discharge, $k$ is the inverse of the mean residence time in the subsurface ($\tau = k^{-1}$), $\alpha$ is the average of precipitation depths and $A$ is the area of the catchment. This corresponds to a Gamma distribution with shape parameter $\lambda/k$ and a scale parameter $\alpha k A$.

The model is suitable for steady state conditions, at the annual or seasonal scale, depending on the temporal variability of the model parameters (Botter et al., 2007a).

Nonlinear storage-discharge relations in subsurface are common due to changes in the hydraulic properties of the soil under different soil moisture levels (Botter et al., 2009; Brutsaert and Nieber, 1977; Mutzner et al., 2013). Accordingly, Botter et al. (2009) proposed an extension of the above modeling framework assuming that:

$$\frac{dQ(t)}{dt} = -k_n Q(t)^a + \xi_t, \tag{5}$$

where $k_n$ and $a$ are the constants of the nonlinear recession.

As previously, with this new storage discharge relation, it is possible to obtain an equation for the pdf of the daily discharges:

$$p(Q, t \to \infty) = C \left\{ \frac{1}{Q^a} \exp\left[ -\frac{Q^{2-a}}{\alpha k_n (2-a)} + \frac{Q^{1-a}\lambda}{k_n (1-a)} \right] \right\}, \tag{6}$$

where $C$ is a normalizing constant (Botter et al., 2009).

According to Castellarin et al. (2013), the proposed method can be classified as a process-based derived distribution method, because it derives a distribution for daily discharges analytically based on underlying catchment properties and some key precipitation characteristics.

## 2.2 Parameter estimation 1: forward estimation

In this work, we use the term "forward parameter" estimation to emphasize that the parameters are estimated directly from observed data before any model is actually applied. Such method is generally used in the context of this model framework for the estimation of the precipitation parameters $\lambda_P$ and $\alpha$. In the present work, the mentioned parameters were always estimated



in a forward mode; however, between the soil-dependent parameters ($\lambda$, $k$, $k_n$ and $a$) the recession parameters ($k$, $k_n$ and $a$) are either estimated in a forward mode (Botter et al., 2007c, 2009; Ceola et al., 2010; Schaefli et al., 2013) or in an inverse mode (Ceola et al., 2010) (see Section 2.3).

Prior to the calculation of the precipitation parameters $\lambda_P$ and $\alpha$, it is necessary to subtract interception losses ($I$) from the

5 observed daily precipitation depths. These losses are in fact evaporated (or sublimated in case of snow) before participating to soil moisture dynamics. Following Rodriguez-Iturbe et al. (1999), previous model applications generally assumed that these losses are accounted for when the frequency of precipitation events is corrected to the frequency of discharge producing events. In view of understanding how the model parameters vary in space, in this work it was decided to treat this interception losses explicitly with minimal assumptions about this process: different maximum interception depths are attributed to four different

land covers (4 mm for forests, 2 mm, for low vegetation, 1 mm for impervious areas, 0 mm for water bodies (Gerrits, 2010)). The catchment-scale maximum interception depth is obtained as the area-average of these values but a minimum interception depth of 1 mm is imposed. This catchment-scale interception depth is subtracted from daily precipitation observations, assuming that at a daily time step, all intercepted water evaporates.

The recession parameter for the linear model is calculated directly from observed daily discharge based on a classical

Brutsaert-Nieber recession analysis (Brutsaert and Nieber, 1977; Biswal and Marani, 2010, 2014; Mutzner et al., 2013), considering, however, only discharges below a certain threshold, fixed to 95%. (see the work of Doulatyari et al. (2015) for a method to estimate this parameter from a geomorphic recession model). The nonlinear recession parameters, $k_n$ and $a$ are also obtained based from a recession analysis, using the same discharge threshold via linear regression of the logarithm of ($-dQ/dt$) versus the logarithm of $Q$, where $a$ is the slope and $k_n$ the intercept. In forward estimation, the frequency of precip-

itation events, $\lambda_P$ is corrected to the discharge-producing frequency $\lambda$ using the following relationship:

$$\overline{\tilde{Q}} = \lambda\alpha, \tag{7}$$

where $\overline{\tilde{Q}}$ is the average of the observed daily discharges $\tilde{Q}$. Estimating $\lambda$ from the above equation rather than directly from the soil properties as in Equation 7, has been shown by Ceola et al. (2010) to provide much better results, and this method is used by the majority of studies since then (e.g. Ceola et al., 2010; Botter et al., 2013; Basso et al., 2015).

**2.3 Parameter estimation 2: inverse estimation**

To objectively compare the potential of different model formulations to capture observed flow-duration curves, the recession parameters for the linear and the nonlinear model are also estimated in a classical inverse estimation mode where the model parameters are obtained by maximizing the likelihood function formulated for the model. For the linear model, the likelihood function is obtained from the model as follows:

$$\mathcal{L}(k|\tilde{\mathbf{Q}}, \alpha, \lambda) = \prod_{j=1}^{N} p(\tilde{Q}_j), \tag{8}$$



where $\tilde{Q}_j, j = 1, ..N$ are the observed daily discharges at all time steps $j$, $p(\tilde{Q}_j)$ is the probability of these observed discharges obtained from the model (Equation 4) and $\mathcal{L}(k|\tilde{Q}, \alpha, \lambda)$ is the likelihood function given observed data and the previously estimated model parameters $\alpha$, $\lambda$. For the nonlinear model, the likelihood $\mathcal{L}(k_n, a|\tilde{Q}, \alpha, \lambda)$ is obtained analogously.

## 2.4 Model evaluation criteria

To objectively compare different models, we propose to use here the Kolmogorov-Smirnov distance between the cdfs corresponding to different models (Ceola et al., 2010; Schaefli et al., 2013), i.e. the maximum difference between the values of the empirical and the modeled cumulative distributions. This comparison of the cdfs overcomes an important limitation inherent in the comparison of analytic pdfs and empirical pdfs: namely the question of the choice of the number of classes for the calculation of the empirical pdf from observed data (i.e. via a so-called frequency polygone Naghettini, 2016). The problem
does not arise for cdfs given their cumulative nature.

Given that the nonlinear model formulation has an additional parameter, the linear and the nonlinear models are also compared based on the Akaike information criterion (Burnham and Anderson, 2004):

$$c^{\mathrm{AIC}} = 2n - \ln(\hat{\mathcal{L}}), \tag{9}$$

where $n$ is the number of parameters of the model and $\ln(\hat{\mathcal{L}})$ is the logarithm of the maximum likelihood function obtained by
maximizing Equation 8.

Based on the above criterion, we measure the relative performance increase from the linear to the nonlinear model as follows:

$$r^{\mathrm{AIC}} = \frac{c_n^{\mathrm{AIC}} - c_l^{\mathrm{AIC}}}{c_l^{\mathrm{AIC}}}, \tag{10}$$

where $c_n{}^{\mathrm{AIC}}$ is the Akaike criterion for the nonlinear model and $c_l^{\mathrm{AIC}}$ for the linear model.

In addition to assessing the performance difference between different models, we propose here to compare the obtained
models to the natural variability of discharge cdfs obtained from observed data. Therefore, an empirical long term cdf is constructed, obtained by ranking the observed data in ascending order and dividing the rank numbers by the total sample size. Furthermore, to assess the natural yearly variability, individual cdfs are constructed for each summer season of each civil year (Vogel and Fennessey, 1994). From these curves, cdf envelopes are obtained based on the maximum and minimum values of discharge for each probability class. A reliable model should yield a cdf contained in these curves and be as close as possible
to the long term cdf.

## 3   Case studies

In Switzerland, an important part of precipitation occurs in form of snow that might either build a seasonal snow cover (accumulation and melt within a single year) or, at elevations roughly beyond 3000m a.m.s.l., accumulate interannually in the





form of glaciers. They represent an important storage that influences the hydrological cycle. In the higher parts of Switzerland, discharges are thus strongly influenced by snow and ice accumulation and melt processes. In addition, this relatively small country (41.285 km$^2$), represents a high variety of hydro-climatic conditions (Weingartner and Aschwanden, 1992; Gonseth et al., 2001).

Accordingly, Switzerland presents three main types of hydrological regimes, namely: pluvial, snow-dominated and glacier (Weingartner and Aschwanden, 1992). The hydrological regime is called pluvial if discharge is driven mostly by rainfall events. Snow-influenced regimes can be found in catchments that show a significant seasonal snow cover, i.e. in catchments with mean elevation above roughly 900m a.m.s.l. In these catchments, solid precipitation accumulates during most of the cold season (winter) and is released in spring when temperature raises above 0°C and snow melts. Depending on the extent and depth

of the snow cover, this snowmelt release can last throughout spring until early summer. For catchments where the cycles of accumulation and melting happen during a single year, the hydrological regime is called snow-dominated. For catchments that have a significant amount of glacier cover, snow is accumulated during the entire year and ice melt sustains high streamflows during the entire summer; resulting in glacier regimes.

The strong seasonality in the Swiss hydrological regimes is illustrated in Figure 2 which shows the joint behavior of discharge

and air temperature during an average year for typical cases of the three main types of regimes (Goldach (GOL), Dischmabach (DIS), Rhône à Gletsch (RHG)); air temperature is shown here as a proxy for snow and evapotranspiration processes. The rainfall-driven Goldach river shows the typical summer low flow resulting from evapotranspiration; the Dischmabach shows a snow regime with high summer flows and the Rhône river a glacier regime, for which summer high flow peaks in the same month as air temperature.

Furthermore, for hydrological purposes, Switzerland is typically divided in three main biogeographical regions: "Plateau and Jura", "Alps" and "South of Alps" (see Figure 3). Jura designates the mountain region in the north of the country, not extending beyond 1679 a.m.s.l. in Switzerland. (i.e. low in comparison to the Alps) and where the predominant hydrological regime is pluvial, locally with important karst effects. It is included here in a single region together with the low lying Plateau area (roughly 400 - 1400m a.m.s.l.). The Alps region includes the highest Swiss summit [i.e. the Dufourspitze, at 4634m

a.m.s.l.] and the source areas of some major European rivers (Rhone, Rhine, Po, Danube) (Huss, 2011). Most snow-dominated catchments are located here. The "South of Alps" is also a mountainous region, but it has a warmer climate and presents higher precipitation.

For the present analysis, all catchments with unperturbed discharges (i.e. minimal anthropogenic influence) that are gauged by the Swiss Federal Office for the Environment (FOEN) (FOEN, 2017) are selected, resulting in 26 study catchments in all

three regions (see Figure 3 and Table 1 for their main characteristics). The corresponding regime classification is given by Weingartner and Aschwanden (1992). 23 of the chosen catchments are considered "hydrological study areas" and have an associated dataset with some key characteristics, such as their land use (Aschwanden, 1996). The provided land use data was used to calculate interception. For the other catchments (i.e. the Areuse, Rhône-Gletsch and Venoge), land use was obtained from spatial data (FOS, 2015). The proportions of land use for each catchment can be found in the supplementary information.





This set of catchments includes fourteen catchments in the Plateau and Jura region, seven in the Alps and five in the South of Alps. The areas of the catchments vary from $1.05 \text{km}^2$ to $377 \text{km}^2$ and their mean elevations from 383m to 1860m a.m.s.l.

Besides observed daily discharge, the model requires basin-scale daily precipitation as input.

Most of the previous applications of the models used precipitation from one or several meteorological stations as input (Botter et al., 2007c, a, 2013; Ceola et al., 2010; Basso et al., 2015; Schaefli et al., 2013). Here it was decided to make use of the relatively new spatial precipitation data set of MeteoSwiss with a resolution of 2,2km and an effective resolution between 15km and 20km and extending back to 1961 (MeteoSwiss, 2014a) which can be assumed to give relatively good estimates of area-averaged precipitation (Paschalis et al., 2014; Addor and Fischer, 2015), even in mountainous areas where there are only few meteorological stations.

In addition, we also used the corresponding spatial temperature data set (MeteoSwiss, 2014b) to support the analysis of parameter seasonality.

## 4 Results

### 4.1 Discharge regimes and parameter seasonality

A key aspect of the different Swiss discharge regimes is their strong seasonality, resulting from evapotranspiration and the accumulation and melt of snow and ice. To gain further insights in the hydrological behavior of the different regimes, Figure 4 shows the within-year variability of the model parameters obtained by estimating the parameters in forward mode for moving and overlapping 90-day windows (to estimate the parameters for a given time window, the data points corresponding to this window in all civil years are pooled together). The precipitation parameters $\alpha$ and $\lambda_P$ overall do not show strong seasonal patterns, except for a few catchments such as the Goldach river (Figure 4a). For snow and glacier regimes, the frequency of discharge-producing events, $\lambda$, increases strongly at the beginning of spring (Figure 4b and c), which indicates the release of water from snow- or ice-melt.

The inverse of the recession coefficient $k$ shows a strong coherent annual cycle for all catchments, independent of the underlying discharge regime (Figure 5). This seasonal pattern with consistently low $\tau$ values during summer for all catchments clearly justifies the choice of a common summer season (June, July, August) for all regimes. The annual cycle (the difference between high and low $\tau$ values) is stronger for snow or glacier regime catchments, which reflects the fact that in these regimes, parts of the catchment are effectively dormant during the winter (Schaefli et al., 2013).

### 4.2 Linear model

All estimated parameters for both forward and inverse estimations are summarized in Table 2. It can be noted that for 11 catchments (i.e. Rein da Sumvitg, Dischmabach, Alpbach, Grosstalbach, Rhône à Gletsch, Massa, Verzasca, Riale di Calneggia, Krumbach, Poschiavino and Ova da Cluozza) , $\lambda$ is bigger than $\lambda_P$, contradicting the original description of the model that states that the discharge producing frequency is smaller than the precipitation frequency. This happens only in the catchments



with presence of snow processes, where snow- or glacier melt increases discharge production during the warmer seasons of the year (i.e. summer and spring), surpassing the losses by evapotranspiration that would cause a decreasing of lambda as described by Botter et al. (2007b). Those results are further discussed in Section 5.

The cdfs obtained from these parameters are presented for three cases endowed with different hydrological regimes (see Figure 6). For the catchment with rainfall-driven discharges (GOL) it can be seen that for the forward estimation, probabilities of occurrence of low flows are largely overestimated (Figure 6a), a typical indication that the recession parameter is underestimated. The model values even exceed the envelopes that represent the natural variability of the discharges. Interestingly, in the presence of snow, the linear model tends to underestimate low flows, with satisfactory results for some cases, such as the Dischmabach. When the relative glacier cover increases, the quality of results start to decrease again as a result of the underestimation of low flows and overestimation of higher flows. This pattern is typical for overestimated recession parameters. It is interesting to notice that the values of the performance indicator (the Kolmogorov-Smirnov distance) are similar for the Goldach (GOL, pluvial regime) and the Rhone river (RHG, glacier regime), around 0.19, despite the difference in the nature of the inaccuracy.

The results are not satisfactory for the forward estimation method, not even for the catchments that are pluvial during all year. This originally motivated a deeper study of the recession parameter and its calculation using an inverse method (i.e. via parameter fitting with MLE). Results for the inverse estimation of parameters are also presented in the form of Kolmogorov-Smirnov distances in Table 2.

The inverse estimation of the model parameters improves the results significantly, but they are still not satisfying, the performance indicator has high values and the curves are visually not accurate specially for pluvial regimes. This suggest that the model with a linear discharge decay is overall not suitable for the studied catchments.

## 4.3 Nonlinear models

The nonlinear model formulation leads overall to a significant model performance increase, as can be verified comparing the indicators of performance $c^{KS}$ and $c^{AIC}$ presented for the linear and nonlinear models in Table 2 and also visually in Figure 7. The improvement is more noticeable for the catchments with low mean elevation and consequently rainfall-driven-regimes, as shown in Figure 8 and by comparing Figures 9a and 9b. Such behavior confirms that the recession observed in these catchments decays is in general better described by a nonlinear model. For some catchments (i.e. Murg-Wängi, Gürbe, Sense, Ilfis, and Grosstalbach), the forward estimation method gives very good results with KS distances below 0.1. In general, for the catchments where the discrepancies between modeled and observed cdfs are due to an underestimation of $\tau$, the improvement of the results is very noticeable. For catchments where the recession time scale is overestimated with the linear model, the nonlinear model estimated in forward model leads to a performance decrease. The results obtained from inverse parameter estimation are very good for all catchments, validating the suitability of the nonlinear model for Swiss river regimes. Again, the nature of the discrepancies between modeled and observed cdf is different for the pluvial catchments than for the catchments where there is snow and glacier cover.





## 4.4 Linear versus nonlinear model performance

As can bee seen from the summary of results in Table 2, the nonlinear model outperforms the linear model for all catchments, both in terms of the KS performance and in terms of the Akaike criterion. Comparing the KS indicators to the mean catchment elevation, it can be seen that the performance increases with elevation for the linear model (Figure 9a) and decreases with eleva-

tion for the nonlinear model (Figure 9b). Elevation differences are more pronounced if the parameters are obtained by forward estimation than by inverse estimation. A plot of the relative performance increase $r_{AIC}$ of the nonlinear model with respect to the linear model against mean catchment elevation (see Figure 8) shows that the nonlinear model performs significantly better for low-elevation catchments and that the difference decreases for high-elevation catchments.

## 5  Discussion

Our results for the 26 Swiss catchments show that the analytic flow distribution model presents very good performances for summer even in areas with snow- and glacier melt-driven summer high flows. Since the model parameters are obtained directly from observed data for each region and for each period, a traditional model validation procedure would not be suitable in this situation. The good performances in many different catchments with different regimes can, however, be considered as a validation of the model for Switzerland. The presence of ice melt in some catchments can be clearly seen comparing the

precipitation frequency $\lambda_p$ to the discharge-producing frequency $\lambda$: in fact, for some catchments, this frequency increases instead of decreasing, as it would be expected for rainfall driven regimes outlined in the original description of the model (Botter et al., 2007c). The increase of $\lambda$ can be understood physically as the effect of an extra source of water contributing to the discharge production, in this case, snow- or glacier-melt that happens when temperature raises during warm seasons.

The augmentation of $\lambda$ with respect to $\lambda_p$ increases for catchments where ice melt has more influence on the discharges

reaching values above $1 day^{-1}$ for the few strongly glaciated catchments (Table 2).

In general, considering the forward estimation of parameters, the results for the linear model are better for the catchments with summer high flows, while the results of the nonlinear model are better for catchments with rainfall-driven regimes. The departure of the model from the observed cdfs for catchments with or without a significant snow cover have different origins. For the summer high flows, the model in general underestimates the discharge variability, probably due to an overestimation of

the recession parameter. For the lower catchments, the model in exchange overestimates the variability, resulting in cdfs that exceed the envelopes, probably due to an underestimation of the recession parameter. When the nonlinear model is applied, the recession becomes stronger, improving the results for the pluvial catchments but worsening them for the catchments with summer high flows. This worsening, however, only holds if the model parameters are estimated in forward mode.

A detailed comparison between the performance of the linear and the nonlinear models considering the optimized parameters

obtained from the inverse approach shows that the results for nonlinear model are always better than for the linear model. This underlines that the nonlinear recession suits better the hydrological conditions of all study catchments and pinpoints the need to improve the method to estimate the recession parameters.





Very interesting additional insights can be obtained from the highlighted model performance increase with mean catchment elevation (Figures 9 and 8). This can in fact be explained by the evolution of the regimes with mean catchment elevation, from rainfall-dominated (pluvial) regimes with summer low flow to snowfall-dominated (nival) regimes with summer high flow. This result suggests, in fact, that mean catchment elevation is a very good proxy for this regime shift despite the fact that many other
catchment characteristics vary strongly across the set of studied catchments (area, hypsometric curve, land use etc.). Given the strong link between mean catchment elevation and mean catchment air temperature, this opens very interesting perspectives for parameter regionalization.

In this context, it should be kept in mind that for the present study, $\lambda$ is estimated directly from the relation between discharge and precipitation (see section 2.2). The question of how to estimate this parameter directly from catchment characteristics based
on long term snow cover statistics and data on glacier cover remains to be answered in future work.

## 6   Conclusions

The application of an analytic model framework to 26 Swiss catchments showed that this framework can describe summer discharge distributions across a wide range of discharge regimes, including rainfall-driven regimes with summer flows, but also regimes with snow- and glacier melt-driven summer high flows. The detailed comparison between the performance of
the linear and nonlinear model formulation shows that the description of summer low flows is strongly improved using a nonlinear storage-discharge relationship, whereas the discharge distribution of high summer flows can be well described with a linear relationship. In general, the linear model performance increases for increasing total summer flows or, equivalently, for catchments with higher mean elevation. Future work will focus on regionalizing the model parameters and on extending the model framework to all four seasons for snow-influenced catchments.

*Competing interests.*   The authors declare that they have no conflict of interest.

*Acknowledgements.*   The work of the first author is funded by the Portuguese Science and Technology Foundation (FCT), grant number PD/BD/52663/2014. The work of Bettina Schaefli is funded by the Swiss National Science Foundation (SNSF), grant number PP00P2_157611. The meteorological data were provided by MeteoSwiss (www.meteoswiss.ch) and the discharge data by the Swiss Federal Office for the Environment (2017, www.hydrodaten.admin.ch).



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





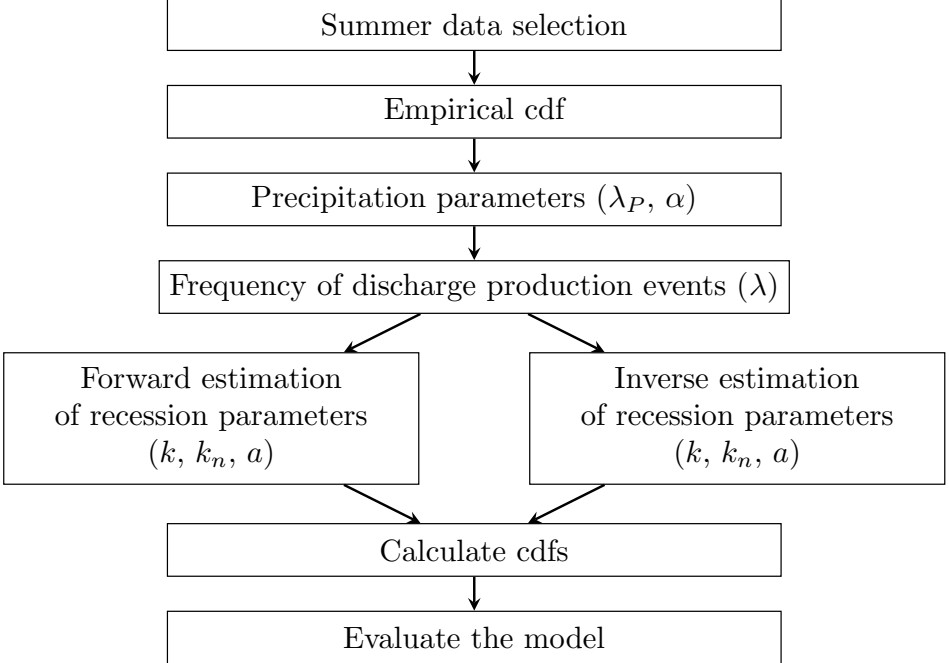

**Figure 1.** Parameter estimation and model evaluation.

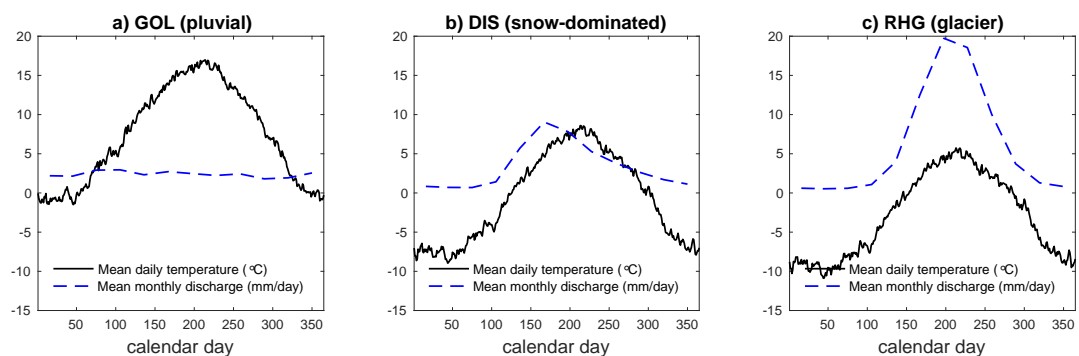

**Figure 2.** Examples of the behavior of discharge and temperature for catchments under different regimes





**Figure 3.** Location of the case studies in Switzerland.





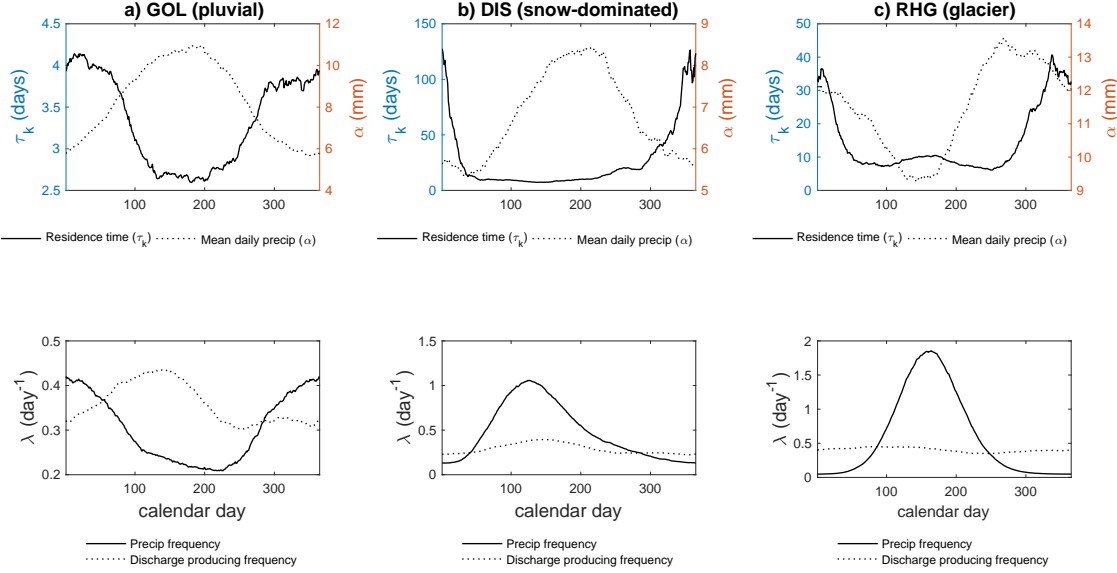

**Figure 4.** Examples of the annual variation of the model parameters. The parameters are calculated for 90 days intervals beginning at the calendar day for which the value is plotted. Top row: residence time $\tau_k$ and mean daily precipitation depth $\alpha$; bottom row: precipitation frequency $\lambda_p$ and discharge-producing frequency $\lambda$.

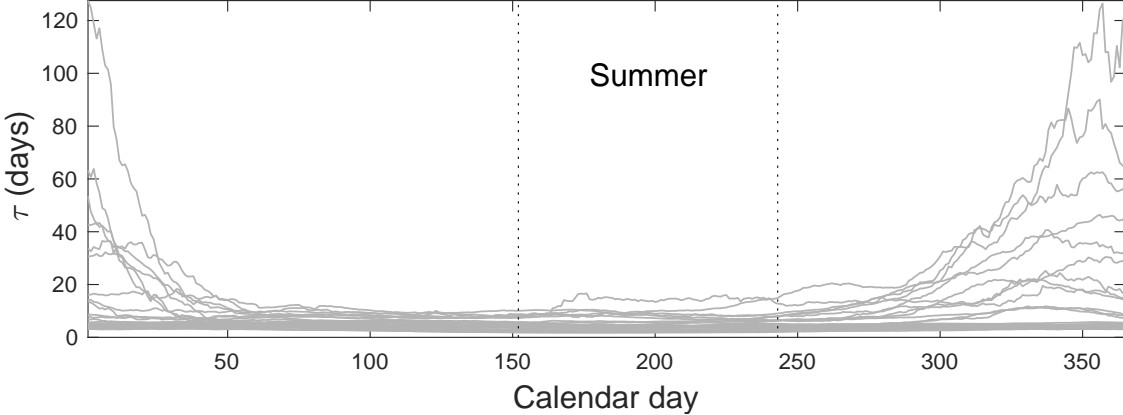

**Figure 5.** Annual variation of the residence time ($\tau_k$) for the 26 catchments.





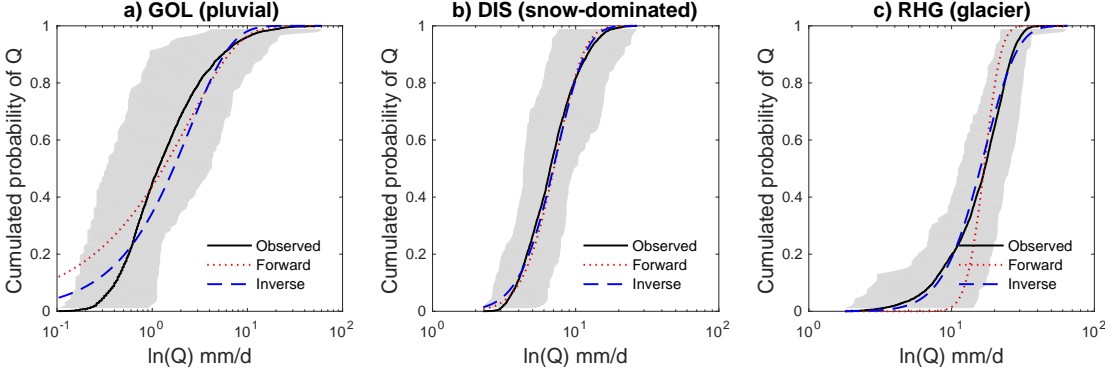

**Figure 6.** Linear model results for summer in the three selected catchments. The area comprehended between the cdf envelopes that represents the natural variability of the discharges is shaded.

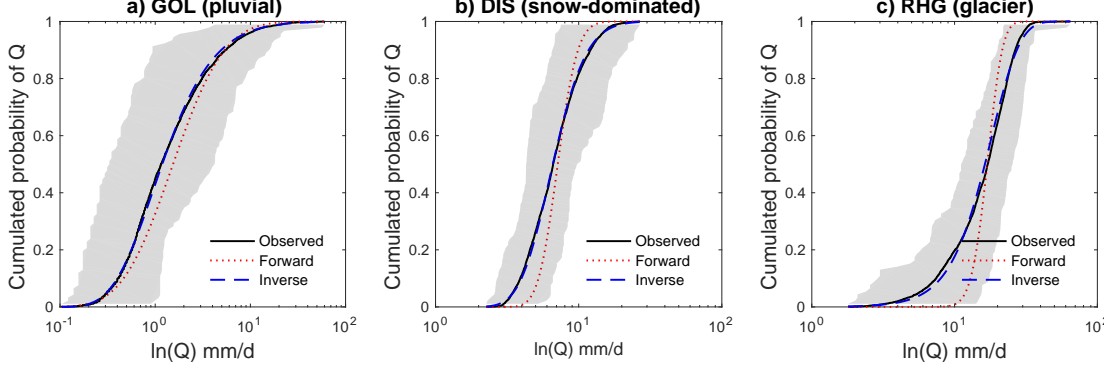

**Figure 7.** Nonlinear model results for summer in selected catchments. The area comprehended between the cdf envelopes that represents the natural variability of the discharges is shaded.




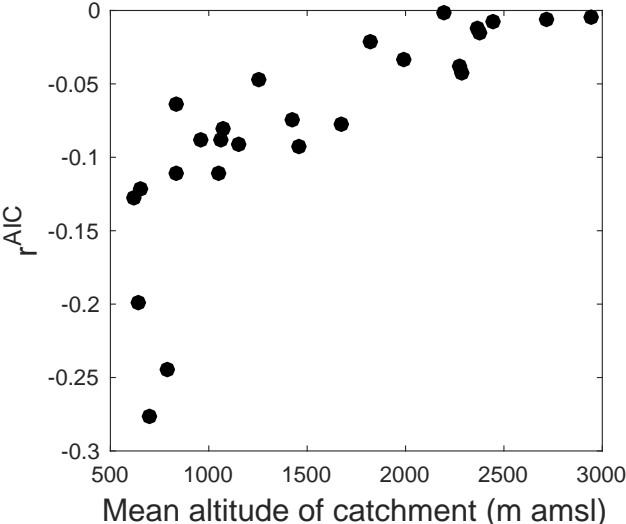

**Figure 8.** Performance of the relative performance increase of the nonlinear with respect to the linear model as a function of mean catchment elevation.

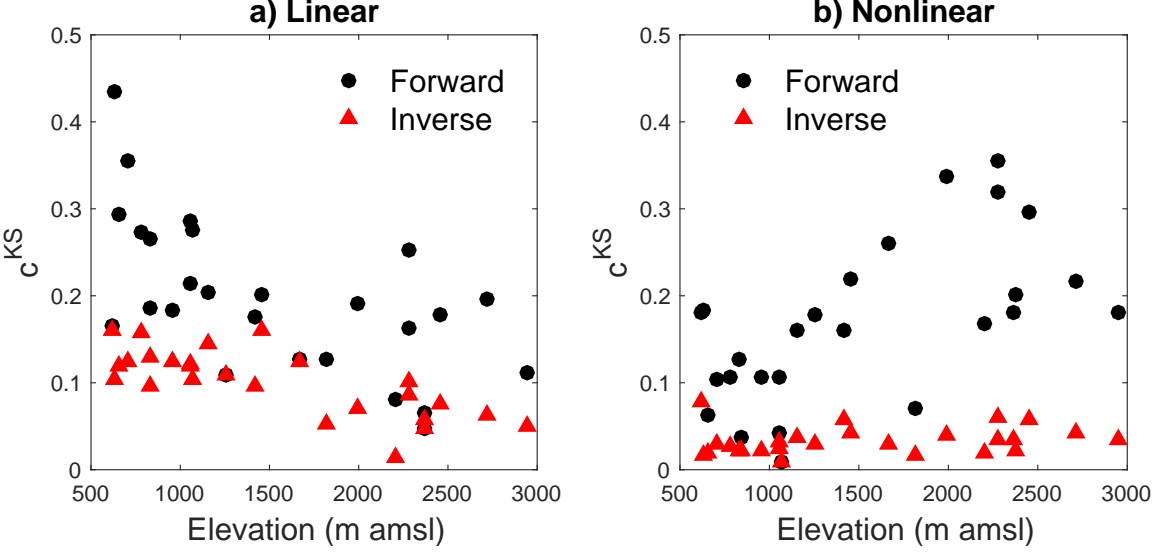

**Figure 9.** Performance of the linear and nonlinear models as a function of mean catchment elevation.



**Table 1.** Characteristics of case studies in Switzerland as given in the FOEN database, including i) the period for which discharge and gridded precipitation are available; ii) regime classification according to the sixteen Swiss regimes and to the simplified three classes used here. P stands for the mean annual precipitation and $\overline{T}$ is the mean annual temperature

| ID | Name | Coordinates | Area (km$^2$) | Mean elevation (m asl.) | Station elevation (m asl.) | Glaciation (%) | P (mm) | $\overline{T}$ ($^o C$) | Data period | Regime 16 classes | Regime 3 classes |
|---|---|---|---|---|---|---|---|---|---|---|---|
| 2430 | Rein da Sumvitg - Sumvitg, Encardens | 718810 / 167690 | 21.8 | 2450 | 1490 | 6.7 | 1707 | -1,19 | 15-09-1977 to 31-12-2014 | b-glacio nival | glacier |
| 2327 | Dischmabach - Davos, Kriegsmatte | 786220 / 183370 | 43.3 | 2372 | 1668 | 2.1 | 1021 | -0,62 | 24-07-1961 to 31-12-2014 | b-glacio nival | glacier |
| 2308 | Goldach - Goldach, Bleiche | 753190 / 261590 | 49.8 | 833 | 399 | 0 | 1446 | 7,39 | 01-01-1974 to 31-12-2014 | pluvial supérieur | pluvial |
| 2374 | Necker - Mogelsberg, Aachsäge | 727110 / 247290 | 88.2 | 959 | 606 | 0 | 1777 | 6,47 | 01-01-1972 to 31-12-2014 | nivo-pluvial préalpin | snow-dominated |
| 2112 | Sitter - Appenzell | 749040 / 244220 | 74.2 | 1252 | 769 | 0.08 | 1904 | 5,10 | 01-01-1961 to 31-12-2014 | nival de transition | snow-dominated |
| 2126 | Murg - Wängi | 714105 / 261720 | 78.9 | 650 | 466 | 0 | 1357 | 7,90 | 01-01-1961 to 31-12-2014 | pluvial inférieur | pluvial |
| 2610 | Scheulte - Vicques | 599485 / 244150 | 72.8 | 785 | 463 | 0 | 1325 | 7,27 | 01-01-1992 to 31-12-2014 | nivo-pluvial jurassien | snow-dominated |
| 2159 | Gürbe - Belp, Mülimatt | 604810 / 192680 | 117 | 837 | 522 | 0 | 1295 | 7,21 | 01-01-1961 to 31-12-2014 | pluvial supérieur | pluvial |
| 2251 | Rotenbach - Plaffeien, Schwyberg | 587980 / 170590 | 1.65 | 1454 | 1275 | 0 | 1910 | 5,81 | 01-09-1961 to 31-12-2014 | nivo-pluvial préalpin | snow-dominated |
| 2179 | Sense - Thörishaus, Sensematt | 593350 / 193020 | 352 | 1068 | 553 | 0 | 1479 | 6,29 | 01-01-1961 to 31-12-2014 | nivo-pluvial préalpin | snow-dominated |
| 2480 | Areuse - Boudry | 554350 / 199940 | 377 | 1060 | 444 | 0 | 1531 | 5,41 | 01-01-1961 to 31-12-2014 | pluvial jurassien | pluvial |
| 2603 | Ilfis - Langnau | 627320 / 198600 | 188 | 1051 | 685 | 0 | 1719 | 6,22 | 01-04-1989 to 31-12-2014 | nivo-pluvial préalpin | snow-dominated |
| 2471 | Murg - Murgenthal, Walliswil | 629340 / 233555 | 207 | 637 | 419 | 0 | 1252 | 7,84 | 26-06-1980 to 31-12-2014 | pluvial inférieur | pluvial |
| 2608 | Sellenbodenbach - Neuenkirch | 658530 / 218290 | 10.5 | 615 | 515 | 0 | 1230 | 8,72 | 12-09-1980 to 31-12-2014 | pluvial inférieur | pluvial |
| 2299 | Alpbach - Erstfeld, Bodenberg | 688560 / 185120 | 20.6 | 2200 | 1022 | 27.7 | 1645 | 0,68 | 01-01-1961 to 31-12-2014 | b-glaciaire | glacier |
| 2276 | Grosstalbach - Isenthal | 685500 / 196050 | 43.9 | 1820 | 767 | 9.3 | 1801 | 2,22 | 01-01-1961 to 31-12-2014 | nival alpin | snow-dominated |
| 2609 | Alp - Einsiedeln | 698640 / 223020 | 46.4 | 1155 | 840 | 0 | 2005 | 5,43 | 27-02-1991 to 31-12-2014 | nivo-pluvial préalpin | snow-dominated |
| 2268 | Rhone - Gletsch | 670810 / 157200 | 38.9 | 2719 | 1761 | 52.2 | 2066 | -2,98 | 01-01-1961 to 31-12-2014 | a-glaciaire | glacier |
| 2161 | Massa - Blatten bei Naters | 643700 / 137290 | 195 | 2945 | 1446 | 65.9 | 2423 | -3,18 | 01-01-1961 to 31-12-2014 | a-glaciaire | glacier |
| 2432 | Venoge - Ecublens, Les Bois | 532040 / 154160 | 231 | 700 | 383 | 0 | 1181 | 9,29 | 01-01-1979 to 31-12-2014 | pluvial jurassien | pluvial |
| 2206 | Melera - Melera (Valle Morobbia) | 726988 / 114670 | 1.05 | 1419 | 944 | 0 | 1716 | 4,74 | 01-01-2005 to 31-12-2014 | nivo-pluvial méridional | snow-dominated |
| 2605 | Verzasca - Lavertezzo, Campiòi | 708420 / 122920 | 186 | 1672 | 490 | 0 | 2051 | 4,37 | 01-09-1989 to 31-12-2014 | nivo-pluvial méridional | snow-dominated |
| 2356 | Riale di Calneggia - Cavergno, Pontit | 684970 / 135960 | 24 | 1996 | 890 | 0 | 1918 | 2,54 | 01-01-1967 to 31-12-2014 | nival méridional | snow-dominated |
| 2244 | Krummbach - Klusmatten | 644500 / 119420 | 19.8 | 2276 | 1795 | 3 | 1475 | 1,92 | 01-01-1995 to 31-12-2014 | nival méridional | snow-dominated |
| 2366 | Poschiavino - La Rösa | 802120 / 142010 | 14.1 | 2283 | 1860 | 0.35 | 1512 | 0,02 | 01-01-1970 to 31-12-2014 | nival méridional | snow-dominated |
| 2319 | Ova da Cluozza - Zernez | 804930 / 174830 | 26.9 | 2368 | 1509 | 2.2 | 963 | -1,36 | 24-07-1961 to 31-12-2014 | nivo glaciaire | snow-dominated |





**Table 2.** Parameter values and performance indicators for all the catchments for summer with linear model and forward estimation, summer linear model and inverse estimation, summer nonlinear model and forward estimation, winter nonlinear model and inverse estimation, winter linear model and forward estimation. $\overline{Q}$ stands for the mean observed discharge, $P_s$ the mean total precipitation during summer, $\overline{T_s}$ the mean temperature during summer, $I$ for interception depth, $c^{KS}$ for the Kolmogorov-Smirnov distance. The indices stand for: $f$ forward estimation, $i$ inverse estimation, $l$ linear model, $n$ nonlinear model.

| Name | $\overline{Q}$ (mm/d) | $P_s$ (mm) | $\overline{T_s}$ ($^oC$) | $\alpha$ (mm/d) | $\lambda_P$ mm | $I$ (1/d) | $\lambda$ (mm) | $k_f$ (1/d) | $c_{lf}^{KS}$ | $k_i$ (1/d) | $c_{li}^{KS}$ | $c_{li}^{AIC}$ |
|---|---|---|---|---|---|---|---|---|---|---|---|---|
| Rein da Sumvitg - Sumvitg, Encardens | 13,8 | 532 | 5,62 | 12,4 | 0,410 | 1,83 | 1,115 | 0,201 | 0,179 | 0,383 | 0,075 | 21550 |
| Dischmabach - Davos, Kriegsmatte | 7,4 | 378 | 6,49 | 8,2 | 0,377 | 2,29 | 0,906 | 0,136 | 0,065 | 0,163 | 0,048 | 22300 |
| Goldach - Goldach, Bleiche | 2,5 | 513 | 15,15 | 11,0 | 0,376 | 3,13 | 0,224 | 0,370 | 0,187 | 0,236 | 0,130 | 13494 |
| Necker - Mogelsberg, Aachsäge | 3,3 | 600 | 14,22 | 12,2 | 0,393 | 3,30 | 0,273 | 0,435 | 0,183 | 0,275 | 0,125 | 16467 |
| Sitter - Appenzell | 5,4 | 648 | 12,30 | 12,5 | 0,433 | 3,06 | 0,427 | 0,393 | 0,109 | 0,308 | 0,108 | 25067 |
| Murg - Wängi | 1,7 | 432 | 16,07 | 9,6 | 0,348 | 3,13 | 0,174 | 0,282 | 0,293 | 0,105 | 0,120 | 13636 |
| Scheulte - Vicques | 1,5 | 388 | 15,10 | 9,1 | 0,312 | 3,46 | 0,162 | 0,264 | 0,274 | 0,133 | 0,158 | 5262 |
| Gürbe - Belp, Mülimatt | 2,1 | 450 | 15,15 | 9,9 | 0,355 | 3,06 | 0,210 | 0,271 | 0,266 | 0,096 | 0,095 | 15070 |
| Rotenbach - Plaffeien, Schwyberg | 4,3 | 616 | 13,29 | 14,0 | 0,378 | 3,16 | 0,309 | 0,550 | 0,202 | 0,339 | 0,161 | 22856 |
| Sense - Thörishaus, Sensematt | 2,2 | 483 | 13,98 | 10,7 | 0,356 | 3,22 | 0,208 | 0,344 | 0,275 | 0,127 | 0,105 | 16401 |
| Areuse - Boudry | 1,7 | 383 | 13,10 | 8,8 | 0,316 | 3,37 | 0,191 | 0,261 | 0,214 | 0,132 | 0,120 | 14013 |
| Ilfis - Langnau | 2,7 | 567 | 13,79 | 12,4 | 0,373 | 3,40 | 0,220 | 0,362 | 0,287 | 0,149 | 0,123 | 8210 |
| Murg - Murgenthal, Walliswil | 1,2 | 389 | 15,99 | 9,0 | 0,323 | 3,12 | 0,135 | 0,230 | 0,435 | 0,033 | 0,105 | 4779 |
| Sellenbodenbach - Neuenkirch | 2,0 | 431 | 16,86 | 9,7 | 0,357 | 2,99 | 0,207 | 0,381 | 0,165 | 0,285 | 0,161 | 6617 |
| Alpbach - Erstfeld, Bodenberg | 16,5 | 457 | 7,29 | 8,9 | 0,477 | 1,28 | 1,858 | 0,171 | 0,081 | 0,276 | 0,014 | 30444 |
| Grosstalbach - Isenthal | 6,0 | 598 | 8,97 | 11,8 | 0,444 | 2,35 | 0,504 | 0,195 | 0,128 | 0,106 | 0,053 | 22256 |
| Alp - Einsiedeln | 4,7 | 687 | 13,03 | 14,1 | 0,415 | 3,40 | 0,335 | 0,521 | 0,204 | 0,318 | 0,144 | 9763 |
| Rhone - Gletsch | 17,1 | 473 | 3,58 | 9,0 | 0,505 | 1,00 | 1,905 | 0,092 | 0,197 | 0,419 | 0,064 | 32412 |
| Massa - Blatten bei Naters | 17,1 | 739 | 3,48 | 13,9 | 0,533 | 1,00 | 1,228 | 0,130 | 0,112 | 0,272 | 0,049 | 32418 |
| Venoge - Ecublens, Les Bois | 0,7 | 298 | 17,39 | 7,9 | 0,268 | 3,14 | 0,090 | 0,194 | 0,355 | 0,056 | 0,124 | 3737 |
| Melera - Melera (Valle Morobbia) | 3,1 | 562 | 12,64 | 18,1 | 0,273 | 3,87 | 0,174 | 0,142 | 0,176 | 0,079 | 0,096 | 2918 |
| Verzasca - Lavertezzo, Campiòi | 6,0 | 581 | 12,03 | 17,9 | 0,313 | 3,00 | 0,333 | 0,287 | 0,127 | 0,294 | 0,125 | 11649 |
| Riale di Calneggia - Cavergno, Pontit | 8,9 | 482 | 9,96 | 13,5 | 0,332 | 2,04 | 0,655 | 0,173 | 0,192 | 0,352 | 0,071 | 25838 |
| Krummbach - Klusmatten | 6,0 | 317 | 9,30 | 9,2 | 0,294 | 2,35 | 0,656 | 0,117 | 0,253 | 0,297 | 0,102 | 8673 |
| Poschiavino - La Rösa | 5,4 | 424 | 7,83 | 11,1 | 0,323 | 2,49 | 0,490 | 0,125 | 0,162 | 0,199 | 0,087 | 19679 |
| Ova da Cluozza - Zernez | 5,2 | 329 | 6,58 | 8,4 | 0,342 | 1,77 | 0,619 | 0,215 | 0,047 | 0,192 | 0,058 | 21954 |

| Name | $k_{nf}$ | $a_f$ | $c_{nf}^{KS}$ | $k_{ni}$ | $a_i$ | $c_{ni}^{KS}$ | $c_{ni}^{AIC}$ |
|---|---|---|---|---|---|---|---|
| Rein da Sumvitg - Sumvitg, Encardens | 0,029 | 1,46 | 0,296 | 0,110 | 1,52 | 0,057 | 21387 |
| Dischmabach - Davos, Kriegsmatte | 0,013 | 1,73 | 0,201 | 0,031 | 1,86 | 0,022 | 21972 |
| Goldach - Goldach, Bleiche | 0,145 | 1,50 | 0,126 | 0,174 | 1,81 | 0,023 | 11990 |
| Necker - Mogelsberg, Aachsäge | 0,125 | 1,63 | 0,107 | 0,156 | 1,81 | 0,023 | 15015 |
| Sitter - Appenzell | 0,066 | 1,69 | 0,179 | 0,115 | 1,76 | 0,029 | 23888 |
| Murg - Wängi | 0,099 | 1,70 | 0,062 | 0,081 | 1,98 | 0,019 | 11978 |
| Scheulte - Vicques | 0,099 | 1,72 | 0,106 | 0,117 | 2,20 | 0,027 | 3978 |
| Gürbe - Belp, Mülimatt | 0,068 | 1,76 | 0,036 | 0,063 | 1,76 | 0,023 | 14108 |
| Rotenbach - Plaffeien, Schwyberg | 0,080 | 1,81 | 0,218 | 0,154 | 1,87 | 0,043 | 20753 |
| Sense - Thörishaus, Sensematt | 0,084 | 1,85 | 0,010 | 0,082 | 1,86 | 0,009 | 15069 |
| Areuse - Boudry | 0,078 | 1,85 | 0,106 | 0,116 | 1,77 | 0,032 | 12785 |
| Ilfis - Langnau | 0,068 | 1,96 | 0,042 | 0,069 | 2,04 | 0,025 | 7303 |
| Murg - Murgenthal, Walliswil | 0,056 | 2,29 | 0,183 | 0,026 | 2,51 | 0,016 | 3831 |
| Sellenbodenbach - Neuenkirch | 0,184 | 1,38 | 0,181 | 0,271 | 1,49 | 0,077 | 5776 |
| Alpbach - Erstfeld, Bodenberg | 0,057 | 1,17 | 0,168 | 0,156 | 1,21 | 0,020 | 30420 |
| Grosstalbach - Isenthal | 0,017 | 1,88 | 0,070 | 0,025 | 1,86 | 0,016 | 21768 |
| Alp - Einsiedeln | 0,089 | 1,76 | 0,160 | 0,110 | 1,97 | 0,036 | 8870 |
| Rhone - Gletsch | 0,107 | 0,87 | 0,216 | 0,897 | 0,70 | 0,043 | 32234 |
| Massa - Blatten bei Naters | 0,052 | 1,13 | 0,181 | 0,585 | 0,70 | 0,034 | 32274 |
| Venoge - Ecublens, Les Bois | 0,119 | 1,65 | 0,103 | 0,104 | 2,00 | 0,030 | 2706 |
| Melera - Melera (Valle Morobbia) | 0,054 | 0,92 | 0,161 | 0,031 | 1,94 | 0,057 | 2702 |
| Verzasca - Lavertezzo, Campiòi | 0,041 | 1,70 | 0,261 | 0,081 | 1,94 | 0,030 | 10738 |
| Riale di Calneggia - Cavergno, Pontit | 0,014 | 1,79 | 0,336 | 0,077 | 1,79 | 0,039 | 24958 |
| Krummbach - Klusmatten | 0,032 | 1,37 | 0,354 | 0,064 | 1,99 | 0,060 | 8345 |
| Poschiavino - La Rösa | 0,014 | 1,75 | 0,318 | 0,042 | 2,05 | 0,035 | 18837 |
| Ova da Cluozza - Zernez | 0,030 | 1,70 | 0,180 | 0,083 | 1,58 | 0,034 | 21673 |