# Peer review of "Analytical flow duration curves for summer streamflow in Switzerland"

_Hydrology and Earth System Sciences, 2017_

## Referee Comment (RC1) · Anonymous Referee #1 · 12 Sep 2017

In this manuscript, the authors apply a well-known stochastic framework in its linear and nonlinear form to 26 catchments in Switzerland. The authors explicitly consider a forward and inverse parameter estimation technique and present the different results between them in detail. Additionally, the performance is assessed with respect to observed discharge. A strong link between catchment elevation and model performance is found for both the linear and nonlinear model version. Overall, the nonlinear version is yielding higher performance.

The manuscript is easy to understand. The manuscript, however, lacks a proper discussion of the results. This can be seen by the fact that the discussion does not contain any reference to previous work, of which exists plenty (these are also mentioned in the introduction). My biggest point of criticism is that the model does not seem to be

applicable to snow-dominated catchments. It lacks the process of snow melt and thus model parameter compensate and behave contradictorily to theory, which is mentioned throughout the manuscript. The manuscript can thus not be published as such. The authors either have to remove these catchments or more interestingly, they have to show how snow melt can be considered in this stochastic framework. The latter avenue would provide a real advance in the research. At the moment, the novelty of the presented work is the application of the inverse parameter estimation to both a linear and nonlinear stochastic framework to estimate streamflow cdfs. The manuscript has to be substantially improved with respect to motivating this point and the discussion has to at least discuss the obtained model performance with respect to previous work.

Please find my major and minor comments below.

**0.1 Major comments**

Abstract

p. 1, l. 9: The conclusions here are not the same as in the conclusions section regarding snowmelt and snowfall onset. As a matter of fact snowfall onset is not mentioned anywhere else in the manuscript. The conclusions are also not deducible from the abstract.

Introduction

p. 1., l. 21f: This statement suggests that this paper will cover to some extent prediction at ungauged basins (PUB), but this is not the case. It is thus misleading. Also the following paragraphs are (p. 2, l. 1ff and p. 2, l. 6 ff) introducing papers for regionalising fdc parameters for PUB, which deviates from the topic of this paper - the suitability of a linear / nonlinear stochastic framework at locations where streamflow observations are available.

p. 2., l. 31ff.: As pointed out correctly by the included references, this model framework

has been applied in a wide range of hydro-climatic regimes. Specifically, the references to Schaefli et al. (2013) is investigating a very similar set of catchments. The difference to the presented study is that Schaefli et al. (2013) only investigated the linear model and not the nonlinear one. This is just briefly mentioned in the introduction (p. 3, l. 11). The value of this study is the comparison to the nonlinear model and the parameter estimation. The introduction should investigate the difference between these two in depth to motivate the topic.

Methods

p. 6, l. 14f: If recession constants are calculated from daily discharge, how does this method help for prediction at ungauged locations?

p. 6, l. 21ff: I do not understand how lambda_p is estimated from equation 7. There is also a contradiction in the description of this equation in p. 6, l. 23.

Case studies

p. 7, l. 27ff: It is not clear to me why snow-dominated catchments are considered in this study. It is clear from equation 2 and 5, that the model is not representing snow melt by temperatures above 0 degree Celsius. These basins should be removed or the model adapted to represent snow melting processes.

Results

p. 9, l. 19ff: The fact that lambda, the frequency of discharge-producing precipitation events, is related to snow melt indicates that the model is not suitable for some catchments, which limits model applicability. It might get the right answer, but for the wrong reason. This is also emphasised by the statement on p. 9, l. 28ff.

Discussion

p. 11, l. 13f: The model has already been applied in swiss catchments in previous work. This should be discussed here.

p. 11, l. 14ff: Has an increase of the discharge-producing frequency over the precipitation frequency been observed in previous work that considered snow-dominated catchments?

p. 11, l. 21ff: The discussion of the performance has to incorporate the results of previous studies. KS distance have also been used previously.

p. 11, l. 24ff: The authors have to present a discussion here why the recession parameter are underestimated, not only stating that they are.

**0.2 Minor comments**

p. 1., l. 3f: "The model paramters are..." This sentence is misleading because the gridded precipitation product is lumped as the input for the model.

p. 4, l. 2: Figure 1 is not presented in detail in the text. It should help the reader to understand the methods better, but is only referenced here.

p. 4, l. 25: it should read "i.e. of".

p. 5, l. 4: "...start to move in the soil..." is ambiguous. It is not clear what the authors mean by this.

p. 5, l. 27ff: I do not understand this sentence.

p. 8, l. 20ff: The paragraph on the description of the biogeographical regions is not much related to the work and should be removed.

p. 10, l. 10ff: Mention here that KS values are shown in Table 2.

p. 11, l. 6ff: The plot for the relative performance increase does not add important information as the improvement for low elevation catchment can already be seen in Figure 9. It should be removed.

---

## Referee Comment (RC2) · Anonymous Referee #2 · 30 Sep 2017

The paper deals with a series of catchments in Switzerland with various geospatial and climatic characteristics. The authors compared the performance of linear against nonlinear stochastic model and found out that nonlinear one outperforms. The general idea would be interesting to the hydrology community, however, it needs to be more developed. Moreover, since the paper is going to be studied by a wide range of people, it is needed to explain some concepts and parameters with more details and better referencing.

The paper is not well-organized and contains technical and language problems which decrease the scientific credibility of this study although technically it has relatively convincible results. I recommend accepting it, revising it with MAJOR revision. There are some points which the authors need to address before publishing this paper:

[Figure]

1- More attention to results and conclusion parts is needed in the abstract.

2- In the introduction part, very old papers are cited which is necessary to show the history of the used method, however, the number of recently published references are very low and they repeated all over the paper. It is strongly suggested to use more updated references in the paper.

3- It is expected to mention more clearly what are the benefits of this method against others which you decided to use it for the current study.

4- It is very important to compare your results with other studies to show all aspects of your findings relative to others. It can reveal the novelty of your work. In the discussion part, there is no comparison of such type. It is strongly suggested to compare the results with similar studies.

5- It is needed to address data sources very clearly and describe exactly how you have used your data.

6- Since not all the readers are familiar with mentioned comparative methods, explain in more details what Kolmogorov-Smirnov and Akaike methods are and try to cite to studies which used the same criteria for comparison among models.

7- Although the results are interesting, but they need more discussion to appropriately describe the new findings.

8- How do you justify if for a specific year, a part of discharge was related to the melt of the ice from the other year precipitation. How is distinguished? Does it have any effect on your results?

There are many points which highly need rephrasing and corrections, mainly grammatical and language issues. Moreover, it can be seen that the used language in some parts is very similar to conversation rather than a scientific text which causes ambiguity in the text. It is strongly recommended to highly take care of this issue.
*** The discussion and conclusion parts need a complete rephrasing. Therefore I am not going to mention them in Line-by-Line comments part.

Please find the Line-by-line Comments (More rephrasing is expected than mentioned comments):

P.1, L. 12-13: Rephrase the sentence, it is better not to use "about" 2 times.

P.1, L. 14-15: Change "rather than of the . . ."

P.2, L. 1: Use exist(s)

P.2, L. 6 is not clear. Rephrase the sentence.

P.2, L. 17: what do you mean by model development time? It needs more explanation.

P.2, L. 23-26: Use 2 references in a single sentence in such a way is a little strange. Moreover, a sentences is copied from other publication. Try to rephrase the sentence.

P.2, L. 27-33: The paragraph contains just a very long sentence. Try to break it down to several sentences.

P.3, L. 3-14: The first two paragraphs needs rephrasing. More academic language is needed.

P.3, L. 17: 41.285km2 change it to 41,285 Km2 (using "." Is not right, moreover, 2 should be in the power."

P.3, L. 17-18: Most of the sentences are like conversations than academic paper language. They need rephrasing.

P.3, L. 20 and P.3, L23-24: The same exact sentence is repeated which shows that the text is not checked carefully before submitting.

P.4, L. 18: What do you mean by "sequence of subsurface inputs"? Explain more.

P.5, L. 26: Remove "In this work".

P.6., L.1: Use "among" instead of "between".

P.6., L.1: Use comma "," after the parenthesis.

P.6, L.17: Rephrase the sentence in parenthesis.

P.6, L.22: above equation is equation number 7. Therefore in the next line what is the correct equation number? It seems that the orders is not right.

P.6, L.23: Use "was" instead of "is".

P.6, L.27: Use "models" instead of "model".

P.7, L.5: What is the purpose of "here" in this sentence?

P.7, L.7: mention what is the limitation?

P.7, L. 19: use "to assess" and remove "we propose here. It needs rephrasing.

P.7, L.24-25: sentence needs rephrasing.

P.7, L.28: for the first time, mention what "a.m.s.l." means

P.8. Last line: briefly mention what are the "supplementary information" and how they can be found?

P.9, L. 14-15: needs rephrasing.

P.9, L. 30-31: rephrase "that states that".

P.10, L. 19: this suggest(S)

P.10, L. 25-26: It is not possible to understand your result.

P.11, L. 3-5: needs rephrasing

TABLES:

Table1.

**HESSD**

1. There are two columns with the names "Regimes" it is not acceptable.

2. It needs rows number in the first column to quickly and easily find out how many catchments are in the table

3. What are the coordinates? It is not possible to extract any information from this column.

Table2.

1. The caption is not appropriate. Try to mention exactly what this table is supposed to describe in an organized pattern.

FIGURES:

1. Figure 2 does not have vertical axis title

2. Figure 4 caption needs more detail. Precipitation frequency is mentioned in the caption but it is not in the figures. Pay attention that a figure, without the text, should be understandable.

3. Figure 6 caption needs rephrasing

4. Figure 9. It is better to show some trends inside the graph.

---

## Author Comment (AC1) · 2 Oct 2017

We would like to thank the referee for her/his comments that will contribute to improve the quality of our manuscript. Below we present our responses to the remarks and issues raised by the referee.

*1.General comments*

*In this manuscript, the authors apply a well-known stochastic framework in its linear and nonlinear form to 26 catchments in Switzerland. The authors explicitly consider a forward and inverse parameter estimation technique and present the different results between them in detail. Additionally, the performance is assessed with respect to observed discharge. A strong link between catchment elevation and model performance*

[Figure]

*is found for both the linear and nonlinear model version. Overall, the nonlinear version is yielding higher performance*

*The manuscript is easy to understand. The manuscript, however, lacks a proper discussion of the results. This can be seen by the fact that the discussion does not contain any reference to previous work, of which exists plenty (these are also mentioned in the introduction).*

R: We agree with the referee that there are previous works using the same framework (they are cited in the introduction of the submitted manuscript) and that we can improve our discussion. Nevertheless, the case studies and databases are different, as well as the used metrics, which limits the options for direct quantitative comparisons. We give hereafter some more details on the planned comparisons (in response to a similar detail comment).

*My biggest point of criticism is that the model does not seem to be applicable to snow-dominated catchments. It lacks the process of snow melt and thus model parameter compensate and behave contradictorily to theory, which is mentioned throughout the manuscript. The manuscript can thus not be published as such. The authors either have to remove these catchments or more interestingly, they have to show how snow melt can be considered in this stochastic framework. The latter avenue would provide a real advance in the research. At the moment, the novelty of the presented work is the application of the inverse parameter estimation to both a linear and nonlinear stochastic framework to estimate streamflow cdfs. The manuscript has to be substantially improved with respect to motivating this point and the discussion has to at least discuss the obtained model performance with respect to previous work.*

R: A priori, the original model framework is not yet supposed to be suitable for snow-dominated catchments during periods of snowmelt (for winter flow, see Schaefli et al., 2013). Originally, the stochastic inputs to discharge production were modeled based on the mean precipitation depth ($\alpha$) and the frequency of precipitation ($\lambda_p$), which are

estimated from observed precipitation and corrected according to losses (i.e. evapo-transpiration) to obtain the frequency of discharge producing events ($\lambda$), which is expected to be smaller than $\lambda_p$. In many snow-dominated catchments, snowmelt happens mostly during spring and summer discharges are essentially rainfall-driven. Nevertheless, we included some catchments with presence of glaciers (where snow and ice melt definitively continue throughout the summer) to test if the framework could work without adaptation also for those cases.

In fact, it is a common assumption in catchment-scale hydrologic modeling (e.g. Schaefli et al., 2005, HESS) that catchment runoff during snowmelt can be modeled with exactly the same functional relationships as during rainfall by simply feeding so-called "equivalent precipitation" into the runoff-generation module, which is composed of rainfall and simulated snow melt. Building on this, it is tempting to think that the analytical framework used here also works for seasons where there is some snowmelt present. However, we did not want to model snowmelt explicitly at this stage since this would add additional parameters to the model. In exchange, we applied the existing framework directly. The presence of snowmelt is not neglected since the discharge producing frequency is estimated from observed discharge.

The results are surprisingly good, even for catchments with glaciers. What we noticed that happened for those cases was an increasing of $\lambda_p$, which is inline with the idea that discharge results from "equivalent precipitation" (rainfall and melt). This was not discussed in detail in the submitted manuscript but we will include a discussion on this in the revised version. What is important to point out here, is that the additional source of water (i.e. snowmelt) is accommodated in the model as an increase in the frequency of inputs and not as an increase of the amount. We will discuss this in detail in the revised version. But the full extension of the model to account for snowmelt inputs explicitly is left for future research.

*2.Specific comments*

*2.1.Major comments*

*Abstract:*

*p1, l. 9: The conclusions here are not the same as in the conclusions section regarding snowmelt and snowfall onset. As a matter of fact snowfall onset is not mentioned anywhere else in the manuscript. The conclusions are also not deducible from the abstract.*

R: The conclusions are going to be extended mentioning our concerns about snowmelt and snowfall, that will be the subject of future studies, as written in the abstract. Also, the abstract will include a sentence about the good results for catchments that present some snow/glacier-melting processes without requiring adaptation of the model.

*Introduction:*

*p. 1., l. 21f: This statement suggests that this paper will cover to some extent prediction at ungauged basins (PUB), but this is not the case. It is thus misleading. Also the following paragraphs are (p. 2, l. 1ff and p. 2, l. 6 ff) introducing papers for regionalizing fdc parameters for PUB, which deviates from the topic of this paper - the suitability of a linear / nonlinear stochastic framework at locations where streamflow observations are available*

R: We thank the reviewer for this observation and we the agree that we should have mentioned ungauged basins only as an outlook. The introduction will be modified to give less weight to ungauged catchments.

*p. 2., l. 31ff.: As pointed out correctly by the included references, this model framework has been applied in a wide range of hydro-climatic regimes. Specifically, the references to Schaefli et al. (2013) is investigating a very similar set of catchments. The difference to the presented study is that Schaefli et al. (2013) only investigated the linear model and not the nonlinear one. This is just briefly mentioned in the introduction (p. 3, l. 11).*

*The value of this study is the comparison to the nonlinear model and the parameter*

*estimation. The introduction should investigate the difference between these two in depth to motivate the topic.*

R: The work of Schaefli et al. (2013) suggests an adaptation of the linear model to winter discharges in snow-dominated catchments, while we focused our work on summer discharges, which makes impossible to compare results even for the linear model. We will make this clearer in the revised introduction.

Nevertheless, this previous work motivated us to keep studying the linear model to verify if it could be suitable also for other seasons. Interestingly the results from the linear model are better for higher elevation catchments, where there is more influence of snow in the hydrological process in general.

We will stress the comparison between the linear and the non linear model in the introduction.

*Methods*:

*p. 6, l. 14f: If recession constants are calculated from daily discharge, how does this method help for prediction at ungauged locations?*

R: As correctly noted before, this work does not concern ungauged catchments, so the methods used to obtain recession constants cannot be applied to them. We will adapt the introduction to make it clear that ungauged catchments are not the subject of this work, they are a future concern.

*p. 6, l. 21ff: I do not understand how $\lambda_p$ is estimated from equation 7. There is also a contradiction in the description of this equation in p. 6, l. 23.*

R: Thank you for this observation, $\lambda_p$ is not obtained from Equation 7, but from the precipitation data exclusively, being just the frequency of precipitation events. The method to obtain this parameter will be clarified and the contradiction will be corrected, the text should refer to Equation 3, not 7.

*Case studies:*

*p. 7, l. 27ff: It is not clear to me why snow-dominated catchments are considered in this study. It is clear from equation 2 and 5, that the model is not representing snow melt by temperatures above 0 degree Celsius. These basins should be removed or the model adapted to represent snow melting processes*

R: Please refer to our earlier response to the general comments.

*Results*

*p. 9, l. 19ff: The fact that lambda, the frequency of discharge-producing precipitation events, is related to snow melt indicates that the model is not suitable for some catchments, which limits model applicability. It might get the right answer, but for the wrong reason. This is also emphasized by the statement on p. 9, l. 28ff*

R: The increase of $\lambda$ in relation to $\lambda_p$ can be explained by snow melt. The good results show empirically that the model is able to incorporate snow melt as an increase of the discharge producing frequency (as it incorporates evapotranspiration as a decrease of this frequency, as it can be seen in the lower catchments). We will make this clearer in the revised version.

*Discussion*

*p. 11, l. 13f: The model has already been applied in Swiss catchments in previous work. This should be discussed here.*

R: To our knowledge, the model framework has been applied by Basso et al. (2015) to two of our case studies and by Doulatyari et al. (2017) to one of our cases, in both works, the nonlinear model was adopted. It has also been applied by Schaefli et al (2013), but to winter, making a comparison impossible. For the cases with possible comparison, despite slightly different databases and methods, we will calculate the indicator that we adopted (KS) and present a comparison in our discussion.

*p. 11, l. 14ff: Has an increase of the discharge-producing frequency over the precipitation frequency been observed in previous work that considered snow-dominated catchments?*

R: Most of the previous studies explicitly exclude cases and/or seasons where snow processes could influence discharge production. One study that mentions an overestimation of $\lambda$ during spring and underestimation during winter is the very recent work of Doulatyari et al. (2017), that observes that pattern for a single case, the Sitter at Appenzell (also one or our case studies), and proposes the same explanation as we did. But this type of pattern can be seen in other studies. Basso et al. (2015), for example, despite stating that catchments with snow processes were excluded, presents the same case as Doulatyari et al. (2017) where the overestimation of $\lambda$ can be seen and an additional case where for spring, $\lambda$ is equal to $\lambda_p$, the Thur at Jonschwil. That equality, during spring, in a catchment with mean elevation1030m amsl hints towards an additional source of water, probably snowmelt. This type of comparison is not always possible because many papers report only $\lambda$ and not $\lambda_p$. We will discuss this in more detail in the revised version.

*p. 11, l. 21ff: The discussion of the performance has to incorporate the results of previous studies. KS distance have also been used previously.*

R: Comparison with some of the results of KS obtained by Ceola et al (2010) will also be incorporated in the discussion. Despite having different cases, the magnitudes of the values are similar.

*p. 11, l. 24ff: The authors have to present a discussion here why the recession parameter are underestimated, not only stating that they are.*

R: We are still developing additional procedures to address this issue. We will include an appropriate discussion of our present understanding of this underestimation based on previous works from other authors.
*2.2. Minor comments*

*p. 1., l. 3f: "The model paramters are..." This sentence is misleading because the gridded precipitation product is lumped as the input for the model.*

R: We will be clearer about adopting a spatial average of gridded precipitation and we will correct that.

*p. 4, l. 2: Figure 1 is not presented in detail in the text. It should help the reader to understand the methods better, but is only referenced here.*

R: We will add some additional explanations about Figure 1 to the text.

*p. 4, l. 25: it should read "i.e. of".*

R: We will correct this mistake

*p. 5, l. 4: ". . . start to move in the soil. . . " is ambiguous. It is not clear what the authors mean by this.*

R: We will clarify the discharge producing process in the text.

*p. 5, l. 27ff: I do not understand this sentence.*

R: We will improve the explanation about the forward method.

*p. 8, l. 20ff: The paragraph on the description of the biogeographical regions is not much related to the work and should be removed.*

R: The paragraph aimed to show the variety of hydrologic conditions in Switzerland and to introduce the classification of regimes proposed the Federal Office for the Environment presented in Table 1, but we will shorten this description.

*p. 10, l. 10ff: Mention here that KS values are shown in Table 2.*

R: We will mention that the values of the indicator are presented In Table 2.

*p. 11, l. 6ff: The plot for the relative performance increase does not add important*

*information as the improvement for low elevation catchment can already be seen in Figure 9. It should be removed.*

R: We agree that both plots present similar results, but we believe that Figure 9 is clearer. Nevertheless it can be omitted.

References:

- Basso, S., Schirmer, M., and Botter, G.: On the emergence of heavy-tailed streamflow distributions, Advances in Water Resources, 82,98–105, doi:10.1016/j.advwatres.2015.04.013, 2015.

- Ceola, S., Botter, G., Bertuzzo, E., Porporato, A., Rodriguez-Iturbe, I., and Rinaldo, A.: Comparative study of ecohydrological streamflow probability distributions, Water Resources Research, 46, doi:10.1029/2010wr009102, 2010.

- B. Doulatyari, A. Betterle, D.Radny, E. A. Celegon, P. Fanton, M. Schirmer, and G. Botter. Patterns of streamflow regimes along the river network: The case of the thur river. Environ- mental Modelling Software, 93:42–58, jul 2017. doi: 10.1016/j.envsoft.2017.03.002.

- Schaefli, B., Rinaldo, A., and Botter, G.: Analytic probability distributions for snow-dominated streamflow, Water Resources Research, 49, 2701–2713, doi:10.1002/wrcr.20234, 2013.

- Schaefli, B., Hingray, B., Niggli, M., and Musy, A.: A conceptual glacio-hydrological model for high mountainous catchments, Hydrol. Earth Syst. Sci., 9, 95-109, 2005.

---

## Author Comment (AC2) · 20 Oct 2017

We noticed that the legend of Figure 4 (bottom row) was inverted. Enclosed the correction version of that figure.
* * *
[Figure]

[Figure]

Figure 4. Examples of the annual variation of the model parameters. The parameters are calculated for 90 days intervals be-
5   ginning at the calendar day for which the value is plotted. Top row: residence time $\tau_k$ and mean daily precipitation depth $\alpha$;
bottom row: precipitation frequency $\lambda_p$ and discharge-producing frequency $\lambda$. The plots have different y-axis scales.

3

**Fig. 1.**

---

## Author Comment (AC3) · 27 Oct 2017

We would like to thank the referee for her/his comments that will contribute to improve the quality of our manuscript. Below we present our responses to the remarks and issues raised by the referee.

*1- The paper deals with a series of catchments in Switzerland with various geospatial and climatic characteristics. The authors compared the performance of linear against nonlinear stochastic model and found out that nonlinear one outperforms. The general idea would be interesting to the hydrology community, however, it needs to be more developed. Moreover, since the paper is going to be studied by a wide range of people, it is needed to explain some concepts and parameters with more details and better*

[Figure]

*referencing.*

Thanks for the overall positive assessment. We will improve the referencing according to our detailed responses hereafter.

*2- The paper is not well-organized and contains technical and language problems which decrease the scientific credibility of this study although technically it has relatively convincible results.*

Thanks for pointing out that the readability of the manuscript should be improved. The language will be carefully revised. We are not entirely sure what the referee refers to in terms of "technical problems" since the detailed comments essentially mention language problems and text formatting problems. Furthermore, we believe that our original manuscript was well organized.

*3- I recommend accepting it, revising it with MAJOR revision. There are some points which the authors need to address before publishing this paper:*

*4-More attention to results and conclusion parts is needed in the abstract.*

R: We agree that the abstract does not reflect exactly our conclusions and it will be adapted in order to be more consistent with discussion and conclusions. The revised abstract will in particular include the findings about the applicability of the model to conditions of snow melt and exclude the statement about snowfall and snowmelt onsets, that were not treated in the paper. We will also mention that we adopted a forward and an inverse mode to estimate recession parameters.

*5- In the introduction part, very old papers are cited which is necessary to show the history of the used method, however, the number of recently published references are very low and they repeated all over the paper. It is strongly suggested to use more updated references in the paper.*

R: The key references are indeed cited throughout the paper, since they are fundamental for the topic The model of Botter et al. (2007) has been applied in more recent

studies that are not directly relevant for the work at hand, but for completeness we will include them in the literature review (Doulatyari et al., 2017; Muller and Thompson, 2016; Mejia et al., 2014; Pumo et al., 2013; Muneepeerakul et al., 2010).

Muller and Thompson (2016) nicely discusses the usefulness of statistical versus process-based flow-duration curve methods for ungauged catchments and we will update our literature review accordingly. Moreover, Doulatyari et al. (2017) have one case study in common with us and make some brief observations about snow processes that are coherent with our thougts about it, worthening to mention also in our discussion (more details on R7).

*6- It is expected to mention more clearly what are the benefits of this method against others which you decided to use it for the current study.*

R: This model framework has already been tested "for different climatic settings" (Botter et al., 2013; Muller et al., 2014; Pumo et al., 2013), including an application to catchments with strong urbanization (Mejia et al., 2014) and an extension to explicitly account fast flow components (Muneepeerakul et al., 2010). The benefits of a process-based approach (as the one used in our paper) as opposed to purely statistical or empirical methods can be summarized as: i) explicit link of FDC shape to rainfall characteristics and catchment recession characteristics rather than an empirical or statistical link to regional FDC shapes and parameter regionalization from extensive discharge observations; ii) the method is applicable to non-stationary climatic settings thanks to the explicit treatment of rainfall and evapo-transpiration characteristics (Muller and Thompson, 2016):

*7- It is very important to compare your results with other studies to show all aspects of your findings relative to others. It can reveal the novelty of your work. In the discussion part, there is no comparison of such type. It is strongly suggested to compare the results with similar studies.*

R: The novelty of our works is not in terms of proposing a better or new method to

predict FDCs but in a more systematic treatment of the model application (and related parameter estimation and performance assessment) across a range of case studies covering different streamflow regimes. This is the first assessment of this type for this modeling framework, which is very promising to assess and predict river flow regimes across climate ranges (Botter et al., 2013). This systematic analysis is a precondition for its application in ungauged catchments (Muller and Thompson, 2016). This will be made clearer in the revised version.

We agree that our discussion can be extended to compare our work with existing studies on the same catchments. Following our response to a similar question raised by another reviewer, we will include an explicit comparison of our results for the catchments Murg at Wangi and Sitter at Appenzell to the work of Basso et al. (2015) and Doulatyari et al. (2017) (Schaefli et al (2013) studies some of the same cases, but in a different season, winter, making a comparison impossible.) For the above cases, despite slightly different databases and methods, we will calculate the performance indicator that we adopted ($c^{KS}$) and present a comparison in our discussion. Furthermore, this comparison will also focus on the presence of snow processes and how this was dealt with in those studies. We also intend to compare the KS values with the ones obtained by Ceola et al. (2010), which adopted the same performance indicator.

*8- It is needed to address data sources very clearly and describe exactly how you have used your data.*

R: Data sources are addressed in section 3 (Case studies) and we believe that the manuscript was precise about how we dealt with discharge and precipitation data. Discharge data is not public and can be obtained on demand from the Swiss Federal Office for the Environment (on the weblink provided in the references). The codes that officially identify the stations are listed in Table 1, in the column ID, which will be made clearer in the text. We will provide more detail information about the selected precipitation grid cells for each catchment in an excel sheet in the supplementary material, including the exact procedure on how the area-average mean precipitation is

obtained from those grids. Furthermore, we will also upload the used basin contours as a shapefile.

*9- Since not all the readers are familiar with mentioned comparative methods, explain in more details what Kolmogorov-Smirnov and Akaike methods are and try to cite to studies which used the same criteria for comparison among models.*

R: We will include the equation for the Kolmogorov-Smirnov distance, which, besides, was well explained on p. 7, lines 5 following. The Akaike information criterion is a very standard metric to choose between models of different complexity. We will add a comment on this and an additional hydrological references (Laio et al., 2009, Ceola et al.,2010).

*10- Although the results are interesting, but they need more discussion to appropriately describe the new findings.*

R: We will improve the discussion as outlined above (R7). Additionally, we will extend the discussion of the case studies influenced by snow processes (see also R11), in agreement also with a statement of reviewer #1.

*11- How do you justify if for a specific year, a part of discharge was related to the melt of the ice from the other year precipitation. How is distinguished? Does it have any effect on your results?*

R: We do not explicitly address the source additional water (in addition to this summer's rainfall), and accordingly, we do not distinguish between snow melt and icemelt (accumulated during previous years). We answered a similar question in our response to reviewer #1 and transcribe a part of that answer here:

"A priori, the original model framework is not yet supposed to be suitable for snow-dominated catchments during periods of snowmelt (for winter flow, see Schaefli et al., 2013). Originally, the stochastic inputs to discharge production were modeled based on the mean precipitation depth ($\alpha$) and the frequency of precipitation ($\lambda_p$), which are

estimated from observed precipitation and corrected according to losses (i.e. evapo-transpiration) to obtain the frequency of discharge producing events (lambda), which is expected to be smaller than $\lambda_p$. In many snow-dominated catchments, snowmelt happens mostly during spring and summer discharges are essentially rainfall-driven. Nevertheless, we included some catchments with presence of glaciers (where snow and ice melt definitively continue throughout the summer) to test if the framework could work without adaptation also for those cases. In fact, it is a common assumption in catchment-scale hydrologic modeling (e.g. Schaefli et al., 2005, HESS) that catchment runoff during snowmelt can be modeled with exactly the same functional relationships as during rainfall by simply feeding so-called "equivalent precipitation" into the runoff-generation module, which is composed of rainfall and simulated snow melt. Building on this, it is tempting to think that the analytical framework used here also works for seasons where there is some snowmelt present. However, we did not want to model snowmelt explicitly at this stage since this would add additional parameters to the model. In exchange, we applied the existing framework directly. The presence of snowmelt is not neglected since the discharge producing frequency is estimated from observed discharge. The results are surprisingly good, even for catchments with glaciers. What we noticed that happened for those cases was an increasing of $\lambda_p$, which is inline with the idea that discharge results from "equivalent precipitation" (rainfall and melt). This was not discussed in detail in the submitted manuscript but we will include a discussion on this in the revised version. What is important to point out here, is that the additional source of water (i.e. snowmelt) is accommodated in the model as an increase in the frequency of inputs and not as an increase of the amount. We will discuss this in detail in the revised version. But the full extension of the model to account for snowmelt inputs explicitly is left for future research."

*12-There are many points which highly need rephrasing and corrections, mainly grammatical and language issues. Moreover, it can be seen that the used language in some parts is very similar to conversation rather than a scientific text which causes ambiguity in the text. It is strongly recommended to highly take care of this issue.*

We will carefully revise the language throughout the paper.

*13- The discussion and conclusion parts need a complete rephrasing. Therefore I am not going to mention them in Line-by-Line comments part (..)*

R: Thanks for your detailed suggestions. We will address each point in the formal rebuttal to be submitted with the revised version.

*Comments about tables and figures:*

*Table 1.*

*1- There are two columns with the names "Regimes" it is not acceptable.*

R: The columns are named "Regimes 16 classes" and "Regimes 3 classes", we will fit the number of classes in the first line of to make the distinction more evident.

*2- It needs rows number in the first column to quickly and easily find out how many catchments are in the table*

R: We will add a first column, numbering the rows.

*3- What are the coordinates? It is not possible to extract any information from this column.*

R: The coordinates correspond to the Swiss coordinate system (CH1903), that information will be added to the table.

*Table 2.*

*1-The caption is not appropriate. Try to mention exactly what this table is supposed to describe in an organized pattern.*

R: We are going to reorganize the caption in a more objective way.

*FIGURES:*

*1- Figure 2 does not have vertical axis title*

R: We will add both scales (representing temperature and mean monthly discharge) to the graph axes.

*2. Figure 4 caption needs more detail. Precipitation frequency is mentioned in the caption but it is not in the figures. Pay attention that a figure, without the text, should be understandable.*

R: We will organize the figure legends, axes and captions and make everything coherent. We also identified that the legends of the second row (Precip frequency and Discharge producing frequency) are inverted, a mistake that will be corrected.

*3. Figure 6 caption needs rephrasing*

R: We will rephrase the caption of figures 6 and 7.

*4. Figure 9. It is better to show some trends inside the graph.*

R: We will add trends for each graph (linear and nonlinear) and each type of estimation (forward and inverse).

References

- Basso, S., Schirmer, M., and Botter, G.: On the emergence of heavy-tailed streamflow distributions, Advances in Water Resources, 82,98–105, doi:10.1016/j.advwatres.2015.04.013, 2015.

- Botter, G., Basso, S., Rodriguez-Iturbe, I., and Rinaldo, A.: Resilience of river flow regimes, Proc. Natl. Acad. Sci., 110, 12925–12930, 2013.

- Ceola, S., Botter, G., Bertuzzo, E., Porporato, A., Rodriguez-Iturbe, I., and Rinaldo, A.: Comparative study of ecohydrological streamflow probability distributions, Water Resources Research, 46, doi:10.1029/2010wr009102, 2010.

- Doulatyari, B., Betterle, A., Radny, D., Celegon, E. A., Fanton, P., Schirmer, M.,

and Botter, G.: Patterns of streamflow regimes along the river network: The case of the thur river. Environmental Modelling & Software, 93:42–58, jul 2017. doi: 10.1016/j.envsoft.2017.03.002.

- Laio, F., Di Baldassarre, G., and Montanari, A.: Model selection techniques for the frequency analysis of hydrological extremes, Water Resources Research, 45, W07416 10.1029/2007wr006666, 2009.

- Mejia, A., Daly, E., Rossel, F., Jovanovic, T., and Gironas, J.: A stochastic model of streamflow for urbanized basins, Water Resources Research, 50, 1984-2001, 10.1002/2013wr014834, 2014.

- Muller, M. F., Dralle, D. N., and Thompson, S. E.: Analytical model for flow duration curves in seasonally dry climates, Water Resources Research, 50, 5510-5531, 10.1002/2014wr015301, 2014.

- Muller, M. F. and Thompson, S. E.: Comparing statistical and process-based flow duration curve models in ungauged basins and changing rain regimes, Hydrol. Earth Syst. Sci., 20, 669-683, https://doi.org/10.5194/hess-20-669-2016, 2016.

- Muneepeerakul, R., Azaele, S., Botter, G., Rinaldo, A., and Rodriguez-Iturbe, I.: Daily streamflow analysis based on a two-scaled gamma pulse model, Water Resources Research, 46, n/a-n/a, 10.1029/2010wr009286, 2010.

- Pumo, D., Noto, L. V., and Viola, F.: Ecohydrological modelling of flow duration curve in Mediterranean river basins, Advances in Water Resources, 52, 314-327, doi:10.1016/j.advwatres.2012.05.010, 2013

- Schaefli, B., Rinaldo, A., and Botter, G.: Analytic probability distributions for snow-dominated streamflow, Water Resources Research, 49, 2701–2713, doi:10.1002/wrcr.20234, 2013.

- Schaefli, B., Hingray, B., Niggli, M., and Musy, A.: A conceptual glacio-hydrological model for high mountainous catchments, Hydrol. Earth Syst. Sci., 9, 95-109, 2005.
* * *

---

## Author Response (AR1)

**Manuscript No. hess-2017-349 : "Inference on analytical flow duration curves in Swiss alpine environments" by A.C. Santos et al.**

**Response to the comments of the Editor and the Reviewers**

We would like to thank the Editor, Fabrizio Fenicia, for the handling of our paper and all reviewers for their detailed and constructive comments, which helped sharpen the manuscript. We are pleased to submit a new, substantially re-written version of the manuscript, carefully revised along the lines of our responses given in the public discussion. We provide hereafter detailed answers to all points and made a considerable effort to account for them in the revised version of our text. We are also providing a new set of supplementary material, and we plan to also make available the underlying Matlab code if the paper is accepted. We did not yet mention this in the manuscript.

The original comments are given in italic, followed by our detailed responses and an indication of the corresponding changes in the manuscript.

Having answered all the points that were raised, we hope that this revised version is now suitable for publication in HESS.

**Answers to Editor's comments**

*Comments to the Author:*

*The reviewers have provided useful suggestions for revising this paper. In general, I agree with the points raised by the reviewers, which should be carefully addressed in the revised version.*

*Below a few points I did not fully understand:*

*- Equation 2: xi'' represents precipitation that produces runoff, whereas Q is only baseflow, which creates a problem in the water balance equation. Should xi'' be the precipitation that produces baseflow?*

R: In the context of this model framework, what is called "baseflow" in the original paper is all streamflow that is triggered in response to an incoming precipitation event into the subsoil, i.e. all streamflow except direct overland flow. In the original paper (Botter et al., 2007), it is in fact written "Under the above assumptions the steady state pdf of the base flow Q (here identified by the subsurface contribution to runoff) (..)". At other instances, this original paper talks about "subsurface runoff events" .

We made this clearer in the revised version by avoiding the terminology "baseflow". It is now written on p. 2, line 9 ff. of the new version.:

> "One such model is the model developed by Botter et al. (2007c), who derived an analytical description of streamflow distributions as the result of subsurface flow pulses triggered by stochastic rainfall and censored by the soil moisture dynamics."

*- Equation 8: where does Qmodelled enter the likelihood function? Is the calibration parameter only k? This should be specified in the text.*

R: Thanks for this comment. The manuscript was indeed not clear. The daily discharges are not modelled, what is modelled is the probability of discharges. The new version of the manuscript now reads as:

"To objectively compare the potential of different model formulations to capture observed flow-duration curves, the recession parameters for the linear ($k$) and the nonlinear models ($k_n$, $a$) are also estimated in a classical inverse estimation mode where the model parameters are obtained by maximizing the likelihood function formulated for the model. For the linear model, the likelihood function is obtained from the model of Equation 4 as follows:

$$L(k \mid \mathbf{Q}, \boldsymbol{\theta}) = \prod p(Q_j \mid k, \boldsymbol{\theta}) \ ,$$

where the probability $p(.)$ is obtained from equation 4 and $\boldsymbol{\theta} = [\alpha, A, \lambda]$ is the parameter vector containing all parameters that are estimated directly from observed data (i.e. not maximized). For the non-linear case, the likelihood is obtained analogously by replacing $k$ with $k_n$ and $a$ and using $p(.)$ from equation 6.

*- Does the impossibility to validate this model (see discussion) imply that this model can be used only for calibration and not for forecasting?*

R: The objective of the model is the prediction of Flow Duration Curves (FDCs), not forecasting discharges. The difficulty of a traditional validation with split sample testing results from the fact that the meteorological parameters will be different for every period. As it is common in the context of FDC estimation, we consider that the model is validated and suitable for prediction given the good results for a range of different case studies. We added the following sentence to the manuscript (p. 10, line 20 ff. of the new version):

"Overall, the good model performance in many different catchments with different regimes indicates that the modelling framework is suitable for the prediction of FDCs in Switzerland. A more detailed model temporal model validation (e.g. with a split sample test, Klemeš, 1986) is not possible for this framework since the model parameters are obtained directly from observed data for each time period (i.e. they vary from period to period)."

**Answers to Reviewer #1:**

*1.    General comments*

*1.1 In this manuscript, the authors apply a well-known stochastic framework in its linear and nonlinear form to 26 catchments in Switzerland. The authors explicitly consider a forward and inverse parameter estimation technique and present the different results between them in detail. Additionally, the performance is assessed with respect to observed discharge. A strong link between catchment elevation and model performance is found for both the linear and nonlinear model version. Overall, the nonlinear version is yielding higher performance*

*The manuscript is easy to understand.*

Thanks for this positive assessment.

*1.2 The manuscript, however, lacks a proper discussion of the results. This can be seen by the fact that the discussion does not contain any reference to previous work, of which exists plenty (these are also mentioned in the introduction).*

R: Thanks for this comment. We improved the discussion section and namely included a comparison to existing studies (please refer to the new discussion section and further details given in response to the next comment).

*1.3 My biggest point of criticism is that the model does not seem to be applicable to snow-dominated catchments. It lacks the process of snow melt and thus model parameter*

*compensate and behave contradictorily to theory, which is mentioned throughout the manuscript. The manuscript can thus not be published as such. The authors either have to remove these catchments or more interestingly, they have to show how snow melt can be considered in this stochastic framework. The latter avenue would provide a real advance in the research. At the moment, the novelty of the presented work is the application of the inverse parameter estimation to both a linear and nonlinear stochastic framework to estimate streamflow cdfs. The manuscript has to be substantially improved with respect to motivating this point and the discussion has to at least discuss the obtained model performance with respect to previous work.*

R: A priori, the original model framework was not yet supposed to be suitable for snow-dominated catchments during periods of snowmelt (for winter flow, see Schaefli et al., 2013). Originally, the stochastic inputs to discharge production were modeled based on the mean precipitation depth ($\alpha$) and the frequency of precipitation ($\lambda_p$), which are estimated from observed precipitation and corrected according to losses (i.e. evapotranspiration) to obtain the frequency of discharge producing events ($\lambda$), which is expected to be smaller than $\lambda_p$. In many snow-dominated catchments, snowmelt happens mostly during spring; summer discharge is essentially rainfall-driven. Nevertheless, we included some catchments with presence of glaciers (where snow and ice melt definitively continue throughout the summer) to test if the framework could work without adaptation also for those cases.

In fact, it is a common assumption in catchment-scale hydrologic modeling (e.g. Schaefli et al., 2005) that catchment runoff during snowmelt can be modeled with exactly the same functional relationships as during rainfall by simply feeding so-called "equivalent precipitation" into the runoff-generation module, which is composed of rainfall and simulated snow melt. Building on this, it is tempting to think that the analytical framework used here also works for seasons where there is some snowmelt present. However, we did not want to model snowmelt explicitly at this stage since this would add additional parameters to the model. In exchange, we applied the existing framework directly. The presence of snowmelt is not neglected since the discharge producing frequency is estimated from observed discharge.

The results are surprisingly good, even for catchments with glaciers. We noticed however that for those cases $\lambda_p$ increases substantially, which is inline with the idea that discharge results from "equivalent precipitation" (rainfall and melt). What is important to point out here, is that the additional source of water (i.e. snowmelt) is accommodated in the model as an increase in the frequency of precipitation inputs and not as an increase of the amount. This is now discussed in detail in the revised discussion section (p. 10, l. 32 ff.). The extension of the model to account for snowmelt inputs explicitly is left for future research.

*2.  Specific comments*

*2.1. Major comments*

*Abstract:*

*2.1.1 p1, l. 9: The conclusions here are not the same as in the conclusions section regarding snowmelt and snowfall onset. As a matter of fact snowfall onset is not mentioned anywhere else in the manuscript. The conclusions are also not deducible from the abstract.*

R: The abstract was revised and now includes considerations about snowmelt. We also removed the mention of snowfall onset from Section 6, which was confusing. Snowmelt is now explicitly mentioned in the revised conclusions.

*Introduction:*

*2.1.2 p. 1., l. 21f: This statement suggests that this paper will cover to some extent prediction at ungauged basins (PUB), but this is not the case. It is thus misleading. Also the following paragraphs are (p. 2, l. 1ff and p. 2, l. 6 ff) introducing papers for regionalizing fdc parameters for PUB, which deviates from the topic of this paper - the suitability of a linear / nonlinear stochastic framework at locations where streamflow observations are available*

R: Thanks for pointing this out. The mentioned paragraphs with considerations about ungauged catchments were excluded to avoid creating wrong expectations. It is only kept as an outlook in the conclusion section.

*2.1.3 p. 2., l. 31ff.: As pointed out correctly by the included references, this model framework has been applied in a wide range of hydro-climatic regimes. Specifically, the references to Schaefli et al. (2013) is investigating a very similar set of catchments. The difference to the presented study is that Schaefli et al. (2013) only investigated the linear model and not the nonlinear one. This is just briefly mentioned in the introduction (p. 3, l. 11).*

R: The work of Schaefli et al. (2013) suggests an adaptation of the linear model to winter discharges in snow-dominated catchments, while we focused our work on summer discharges, which makes impossible to compare results even for the linear model. This is made clearer in the revised version in page 2, line 24 (see also answer to the next point).

*2.1.4 The value of this study is the comparison to the nonlinear model and the parameter estimation. The introduction should investigate the difference between these two in depth to motivate the topic.*

R: The novelty of the paper is not just studying the nonlinear model but also studying in as far the original model formulation is suitable for summer discharge even in snow-influenced catchments. Interestingly the results from the linear model are better for higher elevation catchments where the influence of snow is stronger. This is now better discussed in the revised discussion section.

We also revised the introduction, which now clearly states the objective of this work. The modified introduction part reads as:

> "The objective of this research is to assess and compare the performance of the model in its linear and its non-linear form for summer flows for a range of Alpine discharge regimes. The selected set of case studies covers all Swiss catchments that have a natural (unperturbed) discharge regime and long term discharge monitoring. Compared to existing studies (e.g. Basso et al., 2015, Ceola et al., 2010, Doulatyari et al.,2017), this paper provides a systematic analysis of all model parameters and of their seasonality, and a comprehensive analysis of a wide range discharge regimes, including namely rainfall-driven and snowfall-influenced regimes. This allows a first detailed view on the suitability of the modeling framework for Alpine summer discharges (influenced by rain and snow) and an assessment of the model performance as a function of the discharge regime." (page 2, lines 30 ff. of the new version)

*Methods:*

*2.1.5 p. 6, l. 14f: If recession constants are calculated from daily discharge, how does this method help for prediction at ungauged locations?*

R: As noted in our previous answer to a similar questions this work does not concern ungauged catchments; the methods used to obtain recession constants cannot be applied to them. The current version of the paper does not mention ungauged catchments anymore.

*2.1.6 p. 6, l. 21ff: I do not understand how lambda_p is estimated from equation 7. There is also a contradiction in the description of this equation in p. 6, l. 23.*

R: Thank you for this observation, which refers to a sentence on how $\lambda$ is obtained from $\lambda_p$. The text was misleading because there was a wrong reference to Equation 7 instead of Equation 3. This has been corrected.

*Case studies:*

*2.1.7 p. 7, l. 27ff: It is not clear to me why snow-dominated catchments are considered in this study. It is clear from equation 2 and 5, that the model is not representing snow melt by temperatures above 0 degree Celsius. These basins should be removed or the model adapted to represent snow melting processes*

R: Please refer to our earlier response to the general comments. Snowmelt is not addressed explicitly, but the behavior of the model in the presence of snowmelt is explained.

*Results*

*2.1.8 p. 9, l. 19ff: The fact that lambda, the frequency of discharge-producing precipitation events, is related to snow melt indicates that the model is not suitable for some catchments, which limits model applicability. It might get the right answer, but for the wrong reason. This is also emphasized by the statement on p. 9, l. 28ff*

R: The increase of $\lambda$ in relation to $\lambda_p$ can be explained by snow melt. The good results show empirically that the model is able to incorporate snow melt as an increase of the discharge producing frequency (as it incorporates evapotranspiration as a decrease of this frequency, as it can be seen in the lower catchments). This issue is now better discussed in the revised Discussion Section.

*Discussion*

*2.1.9 p. 11, l. 13f: The model has already been applied in Swiss catchments in previous work. This should be discussed here.*

R: To our knowledge, the model framework has been applied by Basso et al. (2015) to two of our case studies and by Doulatyari et al. (2017) to one of our case studies; in both works, the nonlinear model was adopted; the linear model was also applied by Basso et al. (2015). (It has also been applied by Schaefli et al (2013), but to winter, making a comparison impossible.) For the cases with possible comparison, despite slightly different databases and methods, we calculated the performance indicator that we adopted (KS) and presented a brief comparison in our revised discussion section.

*2.1.10 p. 11, l. 14ff: Has an increase of the discharge-producing frequency over the precipitation frequency been observed in previous work that considered snow-dominated catchments?*

R: Most of the previous studies explicitly exclude cases and/or seasons where snow processes could influence discharge production. One study that mentions an overestimation of $\lambda$ during spring and an underestimation during winter is the recent work of Doulatyari et al. (2017) (published in the meanwhile), that observes this pattern for a

single case study, the Sitter at Appenzell (also one of our case studies), and proposes the same explanation as we did.

This type of pattern can be seen in the results of other studies, where it is, however, not explicitly discussed. Basso et al. (2015), for example, despite stating that catchments with snow processes were excluded, presents the same Sitter at Appenzell case study as Doulatyari et al. (2017). For another of their case studies (Thur river at Jonschwil, mean elevation 1030m asl.), $\lambda$ equals $\lambda_p$ during spring, which also hints towards an additional source of water, probably snowmelt. This type of comparison is not always possible because many papers report only $\lambda$ and not $\lambda_p$.

The above considerations are added to the revised discussion section.

*2.1.11 p. 11, l. 21ff: The discussion of the performance has to incorporate the results of previous studies. KS distance have also been used previously.*

R: Comparison with some of the results of KS obtained by Ceola et al (2010) are incorporated to the revised discussion (p. 10, l. 23-27). Despite having different case studies, the magnitudes of the values are similar. We also calculated the results for two cases studied by other authors with very similar results (see revised discussion section).

*2.1.12 p. 11, l. 24ff: The authors have to present a discussion here why the recession parameter are underestimated, not only stating that they are.*

R: Thanks for pointing out this omission. We now state in the revised discussion section (p.10, l. 14-15)

"The comparison between the forward and inverse estimation methods shows a clear underestimation of kn for most of the catchments, which was already discussed by Dralle et al. (2015)"

*2.2. Minor comments*

*2.2.1 p. 1., l. 3f: "The model parameters are. . . " This sentence is misleading because the gridded precipitation product is lumped as the input for the model.*

R: Yes, the gridded precipitation is averaged before it is used as a model input. This is now made clear in the revised abstract and in the revised Case studies Section (p. 8, l. 10-12)

*2.2.2 p. 4, l. 2: Figure 1 is not presented in detail in the text. It should help the reader to understand the methods better, but is only referenced here.*

R: We added the following explanations about Figure 1 to the text (p.3, l. 11-16) as follows:

"The model evaluation framework adopted here is synthesized in Figure 1, starting from the empirical cdfs as references for performance evaluation. Next, the precipitation frequency, $\lambda p$ (Section 2.1) is estimated from precipitation and $\lambda$ from observed discharge (Equation 7, Section 2.2). The recession parameters can be obtained in forward mode (Section 2.2) or inverse mode (Section 2.3). Based on these parameters, the model cdf is calculated from the linear model (Equation 4) or the nonlinear model (Equation 6) and its performance is evaluated (Section 2.4)"

*2.2.3 p. 4, l. 25: it should read "i.e. of".*

R: Corrected.

*2.2.4 p. 5, l. 4: ". . . start to move in the soil. . . " is ambiguous. It is not clear what the authors mean by this.*

R: The sentence is rephrased to:

"…$s_l$ is the retention capacity …" (p.4, l, 15)

*2.2.5 p. 5, l. 27ff: I do not understand this sentence.*

The sentence is rephrased to (p. 5, l. 3-4).:

> "We use the term "forward parameter estimation" to emphasize that the parameters are estimated directly from observed data, without calibration."

*2.2.6 p. 8, l. 20ff: The paragraph on the description of the biogeographical regions is not much related to the work and should be removed.*

R: The paragraph aimed to show the variety of hydrologic conditions in Switzerland and to introduce the classification of regimes proposed by the Swiss Federal Office for the Environment (Table 1); the description was shortened (p. 7. l. 18ff.).

*2.2.7 p. 10, l. 10ff: Mention here that KS values are shown in Table 2.*

R: The text is modified to mention that the values of the indicator are presented in Table 2.

*2.2.8 p. 11, l. 6ff: The plot for the relative performance increase does not add important information as the improvement for low elevation catchment can already be seen in Figure 9. It should be removed.*

R: We agree that both plots present similar results, but we believe that Figure 9 is clearer and decided to keep it.

**Answers to Reviewer #2:**

*1- The paper deals with a series of catchments in Switzerland with various geospatial and climatic characteristics. The authors compared the performance of linear against nonlinear stochastic model and found out that nonlinear one outperforms. The general idea would be interesting to the hydrology community, however, it needs to be more developed. Moreover, since the paper is going to be studied by a wide range of people, it is needed to explain some concepts and parameters with more details and better referencing.*

Thanks for the overall positive assessment. We improved the referencing according to our detailed responses hereafter.

*2- The paper is not well-organized and contains technical and language problems which decrease the scientific credibility of this study although technically it has relatively convincible results.*

Thanks for pointing out that the readability of the manuscript should be improved. The language was carefully revised. We are not entirely sure what the reviewer refers to in terms of "technical problems" since the detailed comments essentially mention language problems and text formatting problems. Furthermore, we believe that our original manuscript was well organized.

*3- I recommend accepting it, revising it with MAJOR revision. There are some points which the authors need to address before publishing this paper:*

*4-More attention to results and conclusion parts is needed in the abstract.*

R: We agree that the original abstract did not reflect enough our conclusions and it was adapted in order to be more consistent with the discussion and conclusions. Now it includes the findings about the applicability of the model to conditions of snow melt and excludes the statement about snowfall and snowmelt onsets that were not treated in the paper. It also mentions that we adopted a forward and an inverse mode to estimate recession parameters.

*5- In the introduction part, very old papers are cited which is necessary to show the history of the used method, however, the number of recently published references are very low and they repeated all over the paper. It is strongly suggested to use more updated references in the paper.*

R: The key references are indeed cited throughout the paper, since they are fundamental for the topic. The model of Botter et al. (2007) has been applied in more recent studies, which are now cited. Two of them are very relevant to our paper (Doulatyari et al., 2017, see more details in R.7 and Muller and Thompson, 2016,); others that are not directly relevant for the work at hand.

In particular, Muller and Thompson (2016) nicely discusses the usefulness of statistical versus process-based flow-duration curve methods for ungauged catchments and we updated our literature review accordingly (p.2, l. 17-21.).

*6- It is expected to mention more clearly what are the benefits of this method against others which you decided to use it for the current study.*

R: This model framework has already been tested for different climatic settings (Botter et al., 2013; Muller et al., 2014; Pumo et al., 2013), including an application to catchments with strong urbanization (Mejia et al., 2014) and an extension to explicitly account fast flow components (Muneepeerakul et al., 2010). The benefits of a process-based approach (as the one used in our paper) as opposed to purely statistical or empirical methods can be summarized as: i) explicit link of FDC shape to rainfall characteristics and catchment recession characteristics rather than an empirical or statistical link to regional FDC shapes and parameter regionalization from extensive discharge observations; ii) the method is applicable to non-stationary climatic settings thanks to the explicit treatment of rainfall and evapo-transpiration characteristics (Muller and Thompson, 2016). These benefits are now explicitly stated in the revised introduction section. (page 2, lines 17-21 of the new MS).

*7- It is very important to compare your results with other studies to show all aspects of your findings relative to others. It can reveal the novelty of your work. In the discussion part, there is no comparison of such type. It is strongly suggested to compare the results with similar studies.*

R: This comment is similar to comments of reviewer 1 about the quality of our discussion section. The revised section now better compares our results to previous studies (see also our previous answers to comments 2.1.9, 2.1.10 and 2.1.11). In addition, we would like to emphasize here that the novelty of our work is not in terms of proposing a better or new method to predict FDCs but in a more systematic treatment of the model application (and related parameter estimation and performance assessment) across a range of case studies covering different streamflow regimes. This is the first assessment of this type for this modeling framework, which is very promising to assess and predict river flow regimes across climate ranges (Botter et al., 2013). This systematic analysis is a precondition for its application in ungauged catchments (Muller and Thompson, 2016).

This is now made much clearer in the revised abstract and the revised conclusion.

*8- It is needed to address data sources very clearly and describe exactly how you have used your data.*

R: Data sources are addressed in section 3 (Case studies) and we believe that the manuscript was precise about how we dealt with discharge and precipitation data. Discharge data is not freely downloadable but can be obtained on demand from the Swiss Federal Office for the Environment (on the weblink provided in the references). The codes that officially identify the stations are listed in Table 1, in the column ID, which was made clearer in the text, in the Case studies Section. In the revised version, we provide more detail information about the selected precipitation grid cells for each catchment in an excel sheet in the supplementary material, including the exact procedure on how the area-average mean precipitation is obtained from those grids. Reference to this Supplementary Material is made in the text (p. 8, l.11). Furthermore, we also added the used basin contours as a shapefiles to the Supplementary Material (this is mentioned in the revised case study section).

*9- Since not all the readers are familiar with mentioned comparative methods, explain in more details what Kolmogorov-Smirnov and Akaike methods are and try to cite to studies which used the same criteria for comparison among models*

R: We included the equation for the Kolmogorov-Smirnov distance (p.6, Eq.9), which, was already explained in the original manuscript version (p. 7, lines 11 following). The Akaike information criterion is a very standard metric to choose between models of different complexity. We added a comment on this and additional hydrological references (Laio et al., 2009, Ceola et al.,2010).

*10- Although the results are interesting, but they need more discussion to appropriately describe the new findings.*

R: We improved the discussion as outlined above in response to reviewer 1, comments 2.1.9, 2.1.10 and 2.1.11 (see also R7, reviewer 2). Additionally, we extended the discussion of the case studies influenced by snow processes, in agreement also with a statement of reviewer #1 (see also R11, reviewer 2)

*11- How do you justify if for a specific year, a part of discharge was related to the melt of the ice from the other year precipitation. How is distinguished? Does it have any effect on your results?*

R: We do not explicitly address the additional source of water (in addition to this summer's rainfall), and accordingly, we do not distinguish between snow melt and icemelt (accumulated during previous years). We answered a similar question in our response 1.3 to reviewer #1.

*12-There are many points which highly need rephrasing and corrections, mainly grammatical and language issues. Moreover, it can be seen that the used language in some parts is very similar to conversation rather than a scientific text which causes ambiguity in the text. It is strongly recommended to highly take care of this issue.*

We carefully revised the language throughout the paper.

*13- The discussion and conclusion parts need a complete rephrasing. Therefore I am not going to mention them in Line-by-Line comments part (..)*

We carefully revised both sections.

*P.1, L. 12-13: Rephrase the sentence, it is better not to use "about" 2 times.*

R: The sentence was rephrased .

*P.1, L. 14-15: Change "rather than of the . . ."*

R: The sentence was rephrased.

*P.2, L. 1: Use exist(s)*

R: The sentence was deleted.

*P.2, L. 6 is not clear. Rephrase the sentence.*

R: The sentence was deleted.

*P.2, L. 17: what do you mean by model development time? It needs more explanation.*

R: The sentence is rephrased to:

"Simulation-based methods can provide a detailed description of a hydrological system, but are time consuming and require significant amounts of data"

*P.2, L. 23-26: Use 2 references in a single sentence in such a way is a little strange. Moreover, a sentences is copied from other publication. Try to rephrase the sentence.*

R: The paragraph is rephrased

*P.2, L. 27-33: The paragraph contains just a very long sentence. Try to break it down to several sentences.*

R: The sentence was deleted.

*P.3, L. 3-14: The first two paragraphs needs rephrasing. More academic language is needed.*

R: Both paragraphs were rephrased including new reference and objectives. (page 2, lines 26 ff.).

*P.3, L. 17: 41.285km2 change it to 41,285 Km2 (using "." Is not right, moreover, 2 should be in the power."*

R: The units were corrected in the Case studies Section.

*P.3, L. 17-18: Most of the sentences are like conversations than academic paper language. They need rephrasing.*

R: The sentences were deleted, contents about characteristics of the Swiss catchments were moved to case studies section.

*P.3, L. 20 and P.3, L23-24: The same exact sentence is repeated which shows that the text is not checked carefully before submitting.*

R: The sentences were deleted.

*P.4, L. 18: What do you mean by "sequence of subsurface inputs"? Explain more.*

R: Rephrased as  page 4, lines 4-7):

It is assumed hereby that discharge ($Q$) is the result of a series of precipitation inputs that deliver enough water to fill the water deficit in the soil, i.e. that deliver enough water to raise the soil moisture level above its retention capacity ($\xi_t$)"

*P.5, L. 26: Remove "In this work".*

R: The expression was removed (p. 5, l. 3)

*P.6., L.1: Use "among" instead of "between".*

R: The paragraph as rephrased (p. 5, l. 3-7).

*P.6., L.1: Use comma "," after the parenthesis.*

R: Comma was added (p. 5, l. 5)

*P.6, L.17: Rephrase the sentence in parenthesis.*

R: The sentence deleted.

*P.6, L.22: above equation is equation number 7. Therefore in the next line what is the correct equation number? It seems that the orders is not right.*

R: We should refer to Equation 3. The issue was corrected (p. 5, l. 23)

*P.6, L.23: Use "was" instead of "is".*

R: We changed "is" by "was" (p. 5, l. 25)

*P.6, L.27: Use "models" instead of "model".*

R: The sentence was rephrased (p. 6, l. 3)

*P.7, L.5: What is the purpose of "here" in this sentence?*

R: We excluded the word "here" (p. 6, l. 11)

*P.7, L.7: mention what is the limitation?*

R: The limitation was added to p. 6, l. 18-19, which now reads:

"This comparison of the cdfs overcomes an important limitation inherent in the comparison of analytic pdfs and empirical pdfs. In fact, the choice of the number of classes for the calculation of the empirical pdf from observed data (i.e. via a so-called frequency polygon, Naghettini, 2016) can change the shape of the empirical curve."

*P.7, L. 19: use "to assess" and remove "we propose here. It needs rephrasing.*

R: The sentence was rephrased (p. 7, l. 4)

*P.7, L.24-25: sentence needs rephrasing.*

R: The paragraph was rephrased.

*P.7, L.28: for the first time, mention what "a.m.s.l." means*

R: The meaning was added (p. 7, l. 12).

*P.8. Last line: briefly mention what are the "supplementary information" and how they can be found?*

R: The use of supplementary material is standard. It can be found on the article web page. This does not need explicit text in the manuscript.

*P.9, L. 14-15: needs rephrasing.*

R: The sentence was deleted.

*P.9, L. 30-31: rephrase "that states that".*

R: The sentence is rephrased (now in p. 9, l. 5).

*P.10, L. 19: this suggest(S)*

R: We added the "s" to suggest (p. 9, l. 19).

*P.10, L. 25-26: It is not possible to understand your result. P.11, L. 3-5: needs rephrasing*

R: Sorry, the original text was indeed erroneous and difficult to understand (Such behavior confirms that the recession observed in these catchments decays is in general better described by a nonlinear model). It now reads as:

> "The improvement is more noticeable for catchments with low mean elevation (i.e. for catchments with rainfall-driven-regimes) (Figure 9 and 10). This confirms that the recessions of these catchments are generally better described by a nonlinear model.

*Comments about tables and figures:*

*Table 1.*

*1- There are two columns with the names ``Regimes'' it is not acceptable.*

R: The columns are named ``Regimes 16 classes'' and ``Regimes 3 classes'', we fitted the number in the first line of the column name to make the distinction more evident.

*2- It needs rows number in the first column to quickly and easily find out how many catchments are in the table*

R: We added a first column, numbering the rows.

*3- What are the coordinates? It is not possible to extract any information from this column.*

R: The coordinates correspond to the Swiss coordinate system (CH1903), that information was added to the table.

*Table 2.*

*1-The caption is not appropriate. Try to mention exactly what this table is supposed to describe in an organized pattern.*

R: The caption was rewritten in a more objective way.

*FIGURES:*

*1- Figure 2 does not have vertical axis title*

R: We added both scales (representing temperature and mean monthly discharge) to the graph axes.

*2. Figure 4 caption needs more detail. Precipitation frequency is mentioned in the caption but it is not in the figures. Pay attention that a figure, without the text, should be understandable.*

R: Figure 4 was corrected.

*3. Figure 6 caption needs rephrasing*

R: The caption was rephrased.

*4. Figure 9. It is better to show some trends inside the graph.*

R: We added trends for each graph (linear and nonlinear) and each type of estimation (forward and inverse).

**Additional changes**

We made the following two additional important changes:

- We realized thanks to the repeated exchanges with the Swiss Federal Office for the Environment that the discharge gauge Murg-Murghenthal is influenced by water abstraction, accordingly, we removed it from the final analysis.
- We added a figure showing the difference between the discharge-generating frequency and the precipitation frequency as a function of elevation. The corresponding text reads as

  > "Given that at higher elevations, there is more snow accumulation (and thus melt), this exceedance of $\lambda$ over $\lambda_p$ increases with mean catchment elevation (Figure 6), the limit of $\lambda = \lambda_p$ being at around 1500 m asl. This important result is further discussed in Section 5."

[Figure]

**Figure 6.** Difference between $\lambda$ and $\lambda_p$ as a function of mean catchment elevation.

- We inverted original figures 8 and 9 since the order of referencing changed in the revised text.
- We took the negative value of the criterion $r^{\text{AIC}}$. Since this relative performance change is computed on criteria that have to be minimized, the original formulation was counter-intuitive (higher performance increase for lower $r^{\text{AIC}}$ values).

**Summary of rewriting**

The following sections have been considerably re-written in terms of content and order of the content: Introduction, Discussion, Conclusion. The results section has been shortened to avoid repetitions with the discussion section.

The language of the Methods and Case study sections has been improved.

Because of this rewriting, submitting a track-changed version is impossible. We would like to emphasize here that beyond the changes made to accommodate the reviewers' comments and the above mentioned additional changes, the content of the manuscript did not change.

**References**

- Basso, S., Schirmer, M., and Botter, G.: On the emergence of heavy-tailed streamflow distributions, Advances in Water Resources, 82, 98–105, doi:10.1016/j.advwatres.2015.04.013, 2015
- Botter, G., Porporato, A., Rodriguez-Iturbe, I., and Rinaldo, A.: Basin-scale soil moisture dynamics and the probabilistic characterization of carrier hydrologic

flows: Slow, leaching-prone components of the hydrologic response, Water Resources Research, 43, n/a–n/a, doi:10.1029/2006WR005043, w02417, 2007c

- Botter, G., Basso, S., Rodriguez-Iturbe, I., and Rinaldo, A.: Resilience of river flow regimes, Proceedings of the National Academy of Sciences, 110, 12 925–12 930, doi:10.1073/pnas.1311920110, 2013.

- Brutsaert, W. and Nieber, J. L.: Regionalized drought flow hydrographs from a mature glaciated plateau, Water Resources Research, 13, 637–643, doi:10.1029/wr013i003p00637, 1977.

- Ceola, S., Botter, G., Bertuzzo, E., Porporato, A., Rodriguez-Iturbe, I., and Rinaldo, A.: Comparative study of ecohydrological streamflow probability distributions, Water Resources Research, 46, doi:10.1029/2010wr009102, 2010.

- Dralle, David, Nathaniel Karst, and Sally E. Thompson. "a, b careful: The challenge of scale invariance for comparative analyses in power law models of the streamflow recession." Geophysical Research Letters 42.21 (2015): 9285-9293.

- Doulatyari, B., Betterle, A., Radny, D., Celegon, E. A., Fanton, P., Schirmer, M., and Botter, G.: Patterns of streamflow regimes along the river network: The case of the Thur river, Environmental Modelling & Software, 93, 42–58, doi:10.1016/j.envsoft.2017.03.002, https://doi.org/10.1016/j.envsoft.2017.03.002, 2017.

- Klemeš, V.: Operational testing of hydrological simulation models, Hydrological Sciences Journal, 31, 13–24, doi:10.1080/02626668609491024, https://doi.org/10.1080/02626668609491024, 1986.

- Laio, F., Baldassarre, G. D., and Montanari, A.: Model selection techniques for the frequency analysis of hydrological extremes, Water Resources Research, 45, doi:10.1029/2007wr006666, https://doi.org/10.1029/2007wr006666, 2009.

- Mejía, Alfonso, et al. "A stochastic model of streamflow for urbanized basins." Water Resources Research 50.3, 2014.

- Müller, M. F. and Thompson, S. E.: Comparing statistical and process-based flow duration curve models in ungauged basins and changing rain regimes, Hydrology and Earth System Sciences, 20, 669–683, doi:10.5194/hess-20-669-2016, https://doi.org/10.5194/hess-20-669-2016, 2016.

- Muneepeerakul, Rachata, et al. "Daily streamflow analysis based on a two-scaled gamma pulse model." Water Resources Research 46.11 (2010).

- Pumo, Dario, Leonardo Valerio Noto, and Francesco Viola. "Ecohydrological modelling of flow duration curve in Mediterranean river basins." Advances in water resources 52, 2013.

- Schaefli, B., Hingray, B., Niggli, M., and Musy, A.: A conceptual glacio-hydrological model for high mountainous catchments, Hydrology and Earth System Sciences, 9, 95–109, 2005.

- Schaefli, B., Rinaldo, A., and Botter, G.: Analytic probability distributions for snow-dominated streamflow, Water Resources Research, 49, 2701–2713, doi:10.1002/wrcr.20234, 2013.

---

## Author Response (AR2)

**Response to minor technical comment :**

We thank the reviewer for this useful comment and added a remark on this in the case study section, where it reads now:

" Besides observed daily discharge, the model requires catchment-scale daily precipitation as input. Most of the previous applications of the models used precipitation from one or several meteorological stations as input (Botter et al., 2007c, a, 2013; Ceola et al., 2010; Basso et al., 2015; Schaefli et al., 2013),**\*Start NEW\*** which is potentially limiting for the model performance since good area-averaged input estimates are critical. Recent progress in spaceborne precipitation observation, and in particular the Global Precipitation Measurement (GPM) mission, potentially offers an interesting new data source for area-averaged precipitation estimates, even in such complex terrain as the Swiss Alps (Gabella et al., 2017), with the drawback of covering only short historical periods. **\*END NEW\***. Here, we use the relatively new spatial precipitation data set of MeteoSwiss with a nominal resolution of 2.2 km and an effective resolution between 15 km and 20 km and extending back to 1961 (MeteoSwiss, 2014a). This data set can be assumed to give relatively good estimates of area-averaged precipitation (Paschalis et al., 2014; Addor and Fischer, 2015), even in mountainous areas where there are only few meteorological stations."

**Reference**

Gabella, M., Speirs, P., Hamann, U., Germann, U., and Berne, A.: Measurement of Precipitation in the Alps Using Dual-Polarization C-Band Ground-Based Radars, the GPM Spaceborne Ku-Band Radar, and Rain Gauges, Remote Sensing, 9, 1147

10.3390/rs9111147, 2017.